# Skip the Equations: Learning Behavior of Personalized Dynamical Systems Directly From Data

**Krzysztof Kacprzyk** [1]   **Julianna Piskorz** [1]   **Mihaela van der Schaar** [1]

## Abstract

While black-box approaches are commonly used for data-driven modeling of dynamical systems, they often obscure a system's underlying behavior and properties, limiting adoption in areas such as medicine and pharmacology. A two-step process of discovering ordinary differential equations (ODEs) and their subsequent mathematical analysis can yield insights into the system's dynamics. However, this analysis may be infeasible for complex equations, and refining the ODE to meet certain behavioral requirements can be challenging. Direct semantic modeling has recently been proposed to address these issues by predicting the system's behavior, such as the trajectory's shape, directly from data, bypassing post-hoc mathematical analysis. In this work, we extend the original instantiation, limited to one-dimensional trajectories and inputs, to accommodate multidimensional trajectories with additional personalization, allowing evolution to depend on auxiliary static features (e.g., patient covariates). In a series of experiments, we show how our approach enables practitioners to integrate prior knowledge, understand the dynamics, ensure desired behaviors, and revise the model when necessary.

## 1. Introduction

**Personalized Dynamical Systems**   In machine learning (ML), modeling dynamical systems is a core activity with extensive applications in fields such as physics (Rudy et al., 2017), biology (Chen et al., 2019), engineering (Brunton & Kutz, 2022), and medicine (Lee et al., 2020). We call a dynamical system personalized when its evolution function depends on certain observable features. It is similar to dynamical systems that depend on a parameter (as stud-

ied, for instance, in bifurcation theory (Blanchard et al., 2012)), but we emphasize the fact that we usually have many such parameters, we can observe them, and they are often linked to personal characteristics of individual people. A primary example of a personalized dynamical system (PDS) is the human body, which exhibits a lot of variability when it comes to drug metabolisms, disease progressions, or treatment effects. Beyond physiological systems, other examples of PDS may involve the atmosphere (e.g., temperature or pollution) based on a particular location or the animal population in a particular environment.

**Discovery of Closed-Form ODEs**   In practice, it is often essential to ensure the model's behavior aligns with domain-specific requirements. For instance, when developing drugs, researchers must verify that the pharmacokinetic model (Mould & Upton, 2012) is biologically plausible (e.g., ensuring non-negativity and eventual decay of drug concentrations). Moreover, dosing recommendations may depend on the drug's maximum concentration and its timing (Han et al., 2018). An effective way to achieve this level of understanding is through the discovery of closed-form ordinary differential equations (ODEs) (Bongard & Lipson, 2007), where an algorithm identifies a concise mathematical representation that can then be analyzed by experts. More formally, a *closed-form* function is a function that can be represented by a mathematical expression consisting of a finite number of variables, constants, arithmetic operations ($+, -, \times, \div$), and some well-known functions such as trigonometric functions. Initial approaches to ODE discovery (Bongard & Lipson, 2007; Schmidt & Lipson, 2009) employed genetic programming (Koza, 1992) to search the space of possible equations. More recently, ODEs are often represented as linear combinations of terms from a prespecified library (Brunton et al., 2016b) which makes the search much more efficient. This approach (generally called SINDy) has been extended to various settings, including implicit equations (Kaheman et al., 2020), equations with control (Brunton et al., 2016a), and partial differential equations (Rudy et al., 2017). The extended discussion about related works can be found in Appendix E.

**Limitations of Two-Step Modeling**   Traditionally, to understand a dynamical system's behavior, an ODE needs to

---

[1] University of Cambridge, Cambridge, UK. Correspondence to: Krzysztof Kacprzyk <kk751@cam.ac.uk>.

*Proceedings of the $42^{nd}$ International Conference on Machine Learning*, Vancouver, Canada. PMLR 267, 2025. Copyright 2025 by the author(s).

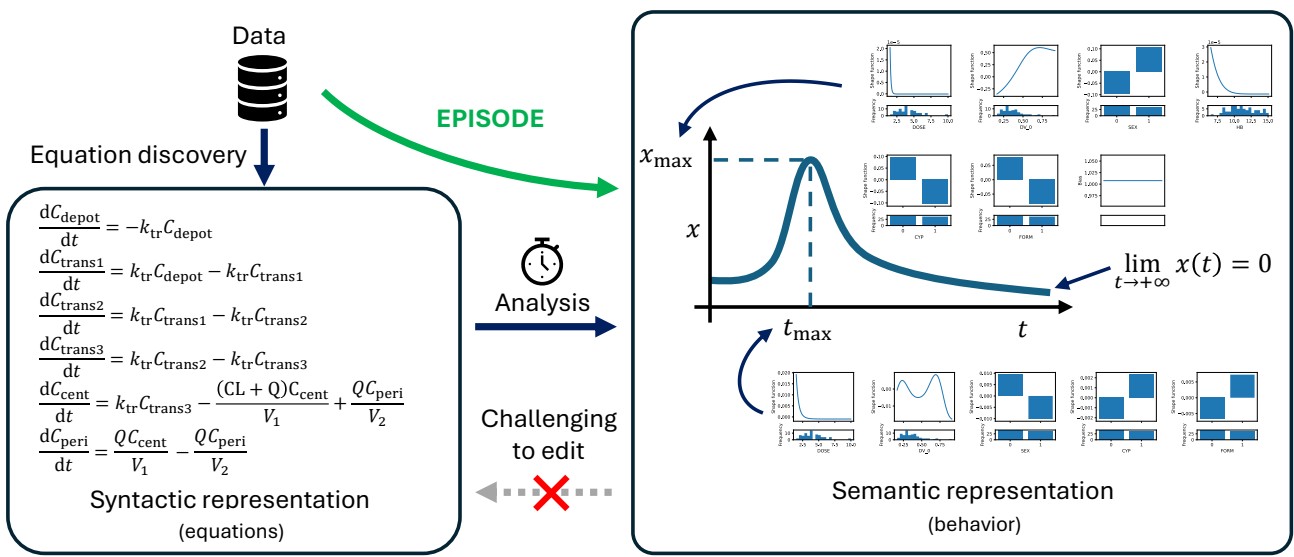

*Figure 1.* In two-step modeling, to learn the behavior of a dynamical system, we need to first discover closed-form governing equations and then perform a time-consuming analysis to arrive at the semantic representation of the model. EPISODE allows us to skip equation discovery and predict the semantic representation directly from data. The figure shows part of EPISODE trained on the Tacrolimus dataset (see Section 5), where we know that the trajectory will always have the shown shape, and its properties, such as the maximum of the trajectory, are described as generalized additive models. Each of the individual shape functions can be scrutinized and edited if necessary, allowing for verification and enforcement of certain conditions, such as $\lim_{t\to\infty} x(t) = 0$.

be first discovered and then analyzed by humans (two-step modeling). This process has many limitations. It is time-intensive, demands substantial mathematical expertise, and can be intractable for complex equations. Moreover, when the discovered ODE fails to meet certain behavioral requirements, revising it is difficult because the link between the form of the equation (its *syntactic representation* and its behavior (its *semantic representation*) is rarely simple.

**Direct Semantic Modeling** To overcome these limitations, *direct semantic modeling* (DSM), has recently been proposed (Kacprzyk & van der Schaar, 2025). It predicts a system's semantic representation (its behavior) directly from data, bypassing post-hoc analysis and enabling greater flexibility. This allows for more intuitive model editing and incorporation of prior knowledge. However, the current instantiation of direct semantic modeling, *Semantic ODE*, is limited to one-dimensional scenarios without personalization (apart from the initial condition).

**Contributions and Outline** In this paper, we develop *EPISODE* (*E*diting and *P*ersonalization *I*n *S*emantic *O*rdinary *D*ifferential *E*quations) that applies the principles of DSM framework to modeling multi-dimensional PDS (Figure 1). In Section 2, we formalize modeling PDS as time series forecasting from static features (extending ODE discovery), describe the principles of DSM, and present the theoretical background necessary for our approach. In Sec-

tion 3, we describe the main ingredients of our model: the *composition map* implemented as a decision tree, and the *property map* constructed from a set of generalized additive models. This complex architecture requires a novel training algorithm, which we describe in Section 4. In Section 5, we present a case study of how EPISODE can discover a novel population pharmacokinetic model with better performance than an expert-designed model from the literature. We demonstrate how our approach enables practitioners to integrate prior knowledge, understand the dynamics, verify the model's properties, and revise it if necessary. Finally, we demonstrate the flexibility of our approach in a series of experiments in Section 6.

## 2. Problem Formalism and Background

In this section, we formulate the problem of time series forecasting from static features which provides a formalism for modeling PDS that generalizes ODEs. Then, we describe the principles of DSM and the theory behind Semantic ODEs on which we build in Section 3.

**Notation and Terminology** Let $t_{\text{end}} \in \mathbb{R}_+ \cup \{+\infty\}$ and $\mathcal{T} = (t_0 = 0, t_{\text{end}})$ be the *time domain*. Let $M \in \mathbb{N}$. We say $\boldsymbol{x} : \mathcal{T} \to \mathbb{R}^M$ is an $M$-dimensional *trajectory* if for each $m \in [M]$ (where $[M] = \{1, \ldots, M\}$), $x_m \in C^2(\mathcal{T})$, i.e., $x_m$ is twice continuously differentiable function on $\mathcal{T}$. We also denote it as $\boldsymbol{x} \in C^2(\mathcal{T})$. We call any ML model whose output is a trajectory a *forecasting model*.

## 2.1. Time Series Forecasting From Static Features

**Definition**   We choose to formulate modeling PDS as time series forecasting from static features. In this setting, we have a dataset $\mathcal{D} = \{\boldsymbol{v}^{(d)}, (t_n^{(d)}, \boldsymbol{y}_n^{(d)})_{n=1}^{N_d}\}_{d=1}^{D}$, where each $\boldsymbol{y}_n^{(d)} \in \mathbb{R}^M$ represents a noisy measurement of some ground truth trajectory $\boldsymbol{x}^{(d)} \in \mathcal{C}^2(\mathcal{T})$ at time point $t_n^{(d)} \in \mathcal{T}$, and $\boldsymbol{v} \in \mathcal{V} \subset \mathbb{R}^K$ are the static features. The goal is to find a forecasting function $F : \mathcal{V} \to C^2(\mathcal{T})$ that predicts an $M$-dimensional trajectory $\boldsymbol{x} : \mathcal{T} \to \mathbb{R}^M$ from the static features $\boldsymbol{v} \in \mathcal{V}$. These features may include things like the patient's covariates, the type and dose of the treatment, the city's location, or initial values of any of the $M$ trajectories.

**Relationship with ODEs**   This setting accommodates and extends the dominant approach to modeling dynamical systems: ODEs. Each system of ODEs $\boldsymbol{f}$ defines a forecasting model $\boldsymbol{F}^{\text{ODE}}$ through the initial value problem (IVP), i.e., for each initial condition $\boldsymbol{x}(0) = \boldsymbol{x}_0 \in \mathbb{R}^M$, $\boldsymbol{F}^{\text{ODE}}$ maps $\boldsymbol{x}_0$ to $M$ trajectories governed by $\boldsymbol{f}$ satisfying this initial condition. Thus, ODE discovery can be treated as a special case of fitting a forecasting model when the static features contain only the initial measurements, i.e., $\boldsymbol{v} = \boldsymbol{x}_0$. Importantly, time series forecasting from static features also allows for trajectories that are *not* governed by fully-observed ODEs or which are only defined for a bounded domain ($t_{\text{end}} \neq +\infty$). It is also useful when we want to predict the whole trajectory without access to the first observation ($\boldsymbol{v}$ does not contain $\boldsymbol{x}_0$), and where we have more variability than can be captured solely by the initial condition.[1]

## 2.2. Direct Semantic Modeling

The discovery of closed-form ODEs is a common way of modeling dynamical systems in a way that potentially allows us to understand the underlying behavior of the system through a careful analysis of the found mathematical equation. However, not only do ODEs not cover all forecasting settings, but the two-step process of first discovering an equation and then analyzing it has many limitations. A framework of DSM has recently been proposed to address these limitations when fitting forecasting models. In our work, we want to apply this framework to the particular problem of time series forecasting from static features. First, we outline the difference between syntactic and semantic representation. Then, we explain the main principle of DSM—prediction using a semantic predictor. Finally, we introduce the definition of semantic representation used in Semantic ODEs, which we use as a starting point in our method.

---

[1]We note, however, that this last point could be accommodated by discovering ODEs *with control* and treating static features as constant control trajectories.

**Forecasting Using Semantic and Trajectory Predictors**
We assume that the main reason for discovering a closed-form ODE is to obtain a model with a concise syntactic representation (symbolic form) that can be subsequently analyzed to obtain the semantic representation of the underlying forecasting model—description of how trajectory's behavior, such as its shape and properties, changes as we vary the input. Recently, DSM has been proposed as an alternative to this two-step process for modeling low-dimensional dynamical systems (Kacprzyk & van der Schaar, 2025). Instead of discovering an equation from data and then analyzing it to obtain its semantic representation, this approach generates the semantic representation directly from data. In general the forecasting model $\boldsymbol{F}$ is defined as a composition $\boldsymbol{F} = \boldsymbol{F}_{\text{traj}} \circ F_{\text{sem}}$. $F_{\text{sem}}$ directly predicts the semantic representation of the trajectory (description of its behaviors), which is passed to the trajectory predictor $\boldsymbol{F}_{\text{traj}}$ that matches this description to trajectories in a given hypothesis space. No post-hoc mathematical analysis is required as the semantic representation of $\boldsymbol{F}$ can be directly accessed through $F_{\text{sem}}$. The model can be easily edited to enforce a specific change in the semantic representation. Incorporating prior knowledge is also streamlined and more intuitive.

### 2.2.1. SEMANTIC REPRESENTATION OF ONE-DIMENSIONAL TRAJECTORY

To realize DSM, we need to formalize the semantic representation of a trajectory that allows us to uniquely match it to an actual trajectory through a trajectory predictor $\boldsymbol{F}_{\text{traj}}$. Semantic ODEs employ such a definition for a one-dimensional trajectory. We will introduce it below as it will form a starting point for our own definition. The semantic representation of trajectory $x : (0, t_{\text{end}}) \to \mathbb{R}$ (note that the original formulation only allows for $t_{\text{end}} = +\infty$, so we extend it here) is described as a pair $(c_x, p_x)$. $c_x$ is the *composition* of $x$, i.e., its overall shape and $p_x$ is the set of properties describing this shape.

**Composition**   More formally, a composition $c_x$ is a sequence of *motifs* where each motif describes the shape of the trajectory on a particular interval, for instance "increasing and strictly convex". We have four *bounded* motifs and six *unbounded* motifs. Each of them is of the form $s_{\pm\pm*}$, i.e., is described by two symbols (each $+$ or $-$) and one letter from $\{b, u, h\}$. The symbols describe the signs of, respectively, the first and second derivatives. The letter $b$ denotes a motif for a **b**ounded time domains, whereas both letter $h$ and $u$ are used for **u**nbounded time domains. The letter $h$ is used for motifs that have a horizontal asymptote. For instance, $s_{+-h}$ is an unbounded motif describing a function that is increasing ($+$), strictly concave ($-$), and with a horizontal asymptote ($h$). An unbounded motif is always the last motif in the composition (if $t_{\text{end}} = +\infty$). We also call such composition *unbounded*. Compositions without an

unbounded motif (i.e., when $t_{\text{end}} \in \mathbb{R}$) are called *bounded*. We call a set of all possible compositions $\mathcal{C}$.

**Properties**   $p_x$ is the set of *properties* describing the composition $c_x$. This includes the coordinates of the *transition points* (points between two consecutive motifs, the point at $t = 0$ and at $t_{\text{end}}$ if $t_{\text{end}} \in \mathbb{R}$). In addition, it also contains information on the first derivative at the first transition point and the first and second derivative at the last transition point. Finally, if $t_{\text{end}} = +\infty$, the set of properties also contains the properties of the unbounded motif. For instance, one of the properties of motif $s_{+-h}$ is the value of the horizontal asymptote. Given a trajectory $x$ with a composition $(s_{--b}, s_{-+h})$, its set of properties is $p_x = (t_0, x(t_0), t_1, x(t_1), \dot{x}(t_0), \dot{x}(t_1), \ddot{x}(t_1), h, t_{1/2})$, where each $(t_i, x(t_i))$ is a transition point, and $(h, t_{1/2})$ are the properties of $s_{-+h}$. We denote the set of all properties for a composition $c$ as $\mathcal{P}_c$ and define $\mathcal{P} = \bigcup_{c \in \mathcal{C}} \mathcal{P}_c$. An example of a trajectory and its semantic representation can be seen in Figure 2.

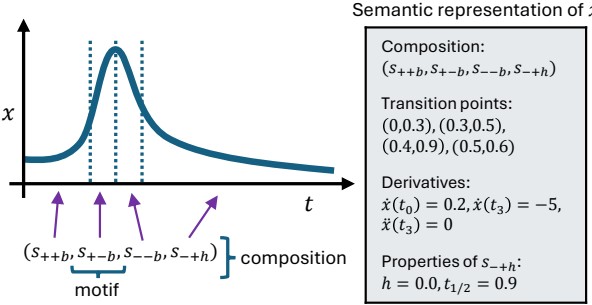

*Figure 2.* An example of a trajectory and its semantic representation. The semantic representation describes the composition of the trajectory, $(s_{++b}, s_{+-b}, s_{--b}, s_{-+h})$, the coordinates of the transition points, some of the derivatives, and properties of the unbounded motif $s_{-+h}$.

# 3. EPISODE

The goal of this work is to realize the DSM framework for the problem of time series forecasting from static features and thus extending the original instantiation, Semantic ODEs, that only works for one-dimensional dynamical systems where the only input is the initial condition (i.e., $M = 1$, $v = x_0$). In a setting where the input is one-dimensional, every property is described by a univariate function that can usually be plotted and understood directly through its graph. For multivariate inputs, the task is much harder as we need to employ and adapt transparent ML models for multivariate regression and classification. We first discuss the high-level structure of our model and then describe its two main ingredients: multivariate composition and property maps.

**Each Dimension Separately**   We propose to treat each dimension of the trajectory separately. As all information needed to predict the whole $M$-dimensional trajectory is encoded in the static features $\boldsymbol{v}$, in particular $\boldsymbol{v}$ is sufficient to predict any of the $M$ dimensions. Our model does not need to take the current values of the remaining dimensions as input, which significantly simplifies the modeling (and understanding) process. That means that our forecasting model $\boldsymbol{F} = \boldsymbol{F}_{\text{traj}} \circ F_{\text{sem}}$ consists of $M$ submodels $\{F_m\}_{m=1}^{M}$ each taking $\boldsymbol{v} \in \mathcal{V}$ and predicting a univariate trajectory $x : \mathcal{T} \to \mathbb{R}$. Then each $F_m$ can be defined as $F_m = F_{\text{traj}} \circ F_{\text{sem}}^{(m)}$, where $F_{\text{traj}}$ is the original trajectory predictor used in Semantic ODEs and $F_{\text{sem}}^{(m)}$ is the new semantic predictor that takes $\boldsymbol{v} \in \mathcal{V}$ as input and outputs a semantic representation of the $m^{\text{th}}$ dimension of the trajectory, i.e., $(c_{x_m}, p_{x_m})$, where $(c_x, p_x)$ is the semantic representation of a one-dimensional trajectory as defined in Section 2.2.1. As for each trajectory dimension $m \in [M]$, we train a separate semantic predictor $F_{\text{sem}}^{(m)}$. Throughout the rest of this section, we fix $m$ and drop the superscript $m$ to improve readability. All that follows describes only one of $M$ semantic predictors. In contrast to the semantic predictor in Semantic ODEs, each $F_{\text{sem}}$ takes a multidimensional vector $\boldsymbol{v} \in \mathcal{V}$ as input instead of a single number. Thus, the comprehensibility of $F_{\text{sem}}$ is in no way guaranteed and requires careful choice of the underlying architecture. As in Semantic ODEs, we describe $F_{\text{sem}}$ as a pair $(F_{\text{com}}, F_{\text{prop}})$, where the former is a *composition map* and the latter a *property map*. We describe them in the next sections. A block diagram showing how a single semantic predictor works is shown in Appendix C.

## 3.1. Composition Map

The goal of the composition map $F_{\text{com}}$ is to predict one part of the semantic representation of the trajectory, namely the *composition* $c \in \mathcal{C}'$, where $\mathcal{C}' \subset \mathcal{C}$ is a set of all admissible compositions chosen by the user. We can interpret $F_{\text{com}}$ as a classification algorithm that takes the static features $v \in \mathcal{V}$ and outputs one of the classes in $\mathcal{C}$. To make the composition map as transparent as possible, we implement it as a classification decision tree of small depth (Breiman, 2017). This allows us to understand how the shape (composition) of the trajectory depends on the static features. An example of such a composition map can be seen in Figure 3, where we show a trained composition map on the Tumor dataset (details in Appendix D.1). Each internal node is a simple condition checking if one of the features is above or below a certain threshold. The leaves (terminal nodes without children) predict one of the admissible compositions.

## 3.2. Property Map

**Property Sub-Maps**   The goal of the property map $F_{\text{prop}} : \mathcal{V} \to \mathcal{P}$ is to describe the properties of the particular compo-

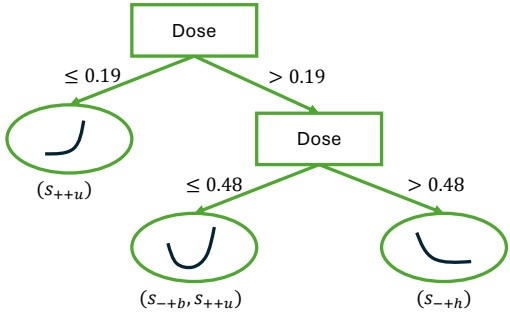

*Figure 3.* Example of a composition map found by our algorithm. Not only did our method find a dose of the drug to be the most important feature determining the trajectory behavior, but it also determined the thresholds where the shape of the trajectory changes. For low doses, the tumor grows exponentially $(s_{++u})$, for higher doses, it decreases at the beginning but then relapses $(s_{-+b}, s_{++u})$, and finally, for very high doses the tumor is predicted to decay $(s_{-+h})$.

sition of the trajectory. As our trajectory may have a different composition depending on the static features $v$ (as predicted by the composition map), the property map consists of a few property sub-maps, each for a different composition predicted by the composition map. For instance, a composition map shown in Figure 3 requires three property sub-maps. We denote them as $F_{\text{prop}}^{s_{++u}}$, $F_{\text{prop}}^{s_{-+b},s_{++u}}$, and $F_{\text{prop}}^{s_{-+h}}$, where each $F_{\text{prop}}^c : \mathcal{V} \to \mathcal{P}_c$. Formally $F_{\text{prop}}(v) = F_{\text{prop}}^{F_{\text{com}}(v)}$.

**Property Functions as GAMs** Each property sub-map $F_{\text{prop}}^c : \mathcal{V} \to \mathcal{P}_c$ is a set of property functions, each predicting a different property from $p_x$. For instance if $c = (s_{-+b}, s_{++u})$, we need 8 property *functions* for each of the property of $p_x = (t_0, x(t_0), t_1, x(t_1), \dot{x}(t_0), \dot{x}(t_1), \ddot{x}(t_1), d)$, where $d$ is the "asymptotic doubling time". To keep these functions transparent, we characterize as many of them as possible as generalized additive models (GAMs) (Hastie & Tibshirani, 1986) or simple transformations of GAMs. GAM is a fully transparent model for a static regression setting where the output is a sum of univariate functions (called shape functions), each taking a different feature. It can be easily comprehended by just plotting all shape functions. As there are no interactions between features, each of the shape functions can be analyzed independently. They proved to be very flexible and transparent models in many real-world settings (Lou et al., 2012; Caruana et al., 2015). Using GAMs, we can, for instance, describe the $x$-coordinate of the second transition point of $(s_{-+b}, s_{++u})$—the minimum of the trajectory—as $x(t_1) = \sum_{k=1}^K g_k(v_k) + g_0$, where $g_0 \in \mathbb{R}$ is a bias term and the individual $g_k$'s can be plotted as seen in Figure 4. Note that when we visualize a GAM, we center each shape function such that its expected value over the marginal distribution is 0. These contributions get absorbed into the bias term, which equals the expected value ($g_0 = \mathbb{E}[x(t_1)]$).

Below each plot, we show the marginal distribution of the corresponding feature in the training dataset. This gives an indication of the confidence in each segment of the function.

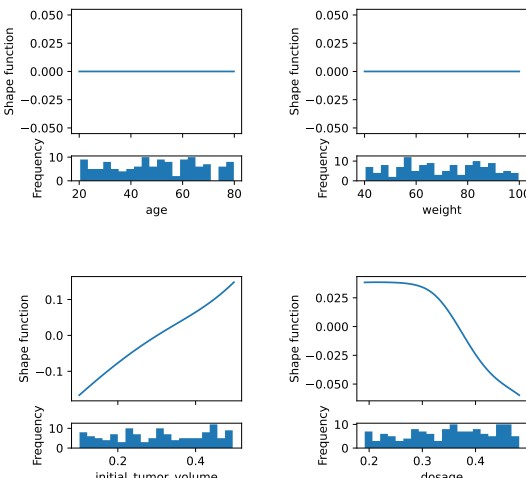

*Figure 4.* Shape functions of a GAM describing $x(t_1)$ of the composition $(s_{-+b}, s_{++u})$ (i.e., the minimum of the trajectory) trained on the Tumor dataset. Each shape function is such that its expected value over the marginal distribution of the feature is 0. The bias (expected value) is equal to 0.212. We see that age and weight do not impact the minimum tumor volume. The other two functions let us observe that the minimum increases linearly with the increase in the initial tumor volume and decreases as the dose of the drug is increased.

## 4. Training

In this section, we describe the training process of a single $F_m = F_{\text{traj}} \circ F_{\text{sem}}^{(m)}$, i.e., a forecasting model that takes $v \in \mathcal{V}$ and outputs a one-dimensional trajectory $x$. We train each $F_m$ separately for all $m \in [M]$. For the remainder of this section, we fix $m$ and drop the superscripts $(m)$ to improve readability. As in Semantic ODEs, the trajectory predictor $F_{\text{traj}}$ is fixed and not trainable. Its job is to match a semantic representation $(c, p) \in \mathcal{C} \times \mathcal{P}$ to a trajectory $x$ such that $(c_x, p_x) = (c, p)$. A block diagram depicting the training procedure of a single semantic predictor is shown in Appendix C.

**Sequential and Parallel Training** As described in Section 3, $F_{\text{sem}} = (F_{com}, F_{prop})$. To train $F_{\text{sem}}$, we first train $F_{\text{com}}$ and then train $F_{\text{prop}}$, i.e., we train each of the property submaps $F_{\text{prop}}^c$ for all the compositions $c$ that are predicted by $F_{\text{com}}$. After the composition map is fitted, all property sub-maps can be trained in parallel.

### 4.1. Training the Composition Map

$F_{\text{com}}$ is a classification decision tree that takes a vector of static features and predicts the composition $c \in \mathcal{C}'$.

**Matching Compositions to Observed Trajectories** We start, similarly to Semantic ODE, by checking how each of the compositions in $\mathcal{C}'$ matches each of our $D$ samples. For each sample $(\boldsymbol{v}^{(d)}, (t_n^{(d)}, y_n^{(d)})_{n=1}^{N_d})$ (where $y_n^{(d)}$ is the $m^{\text{th}}$ coordinate of $\boldsymbol{y}_n^{(d)}$) and composition $c \in \mathcal{C}'$ we calculate $e^{(d)}[c]$ defined as

$$e^{(d)}[c] = \min_{p \in \mathcal{P}_c} \frac{1}{N_d} \sum_{n=1}^{N_d} \left( F_{\text{traj}}(c, p)(t_n^{(d)}) - y_n^{(d)} \right)^2. \quad (1)$$

We search the space $\mathcal{P}_c$ using the L-BFGS algorithm. To improve the fit, before minimizing the mean squared error (MSE) between the $F_{\text{traj}}^0(c, p)$ and $\boldsymbol{y}^{(d)}$, we first minimize the Dynamic Time Warping distance (Müller, 2007). We observed that this procedure performs better, especially when the composition is long.

**Fitting a tree** Fitting a classification decision tree requires a dataset of static features and classification labels. In our setting, instead of a single label for each sample, we have a whole vector of $|\mathcal{C}'|$ real numbers $e^{(d)}$. Instead of minimizing the classification error, we want to minimize the following objective: $\frac{1}{D} \sum_{d=1}^{D} e^{(d)}[F_{\text{com}}(\boldsymbol{v}^{(d)})]$. To find a decision tree that minimizes such an objective, we perform a greedy search where, at each node, we evaluate all possible splits for all possible features (similarly to CART (Breiman, 2017)) and choose the one that gives the smallest error. In practice, we put a few additional constraints to prevent a composition map that is unnecessarily more complex without a significant improvement in performance. We provide more details in Appendix C.3.

### 4.2. Training a Property Sub-Map

Given a composition map $F_{\text{com}}$, we partition our training set depending on the composition predicted by $F_{\text{com}}$. Then for each $c \in \text{Im}(F_{\text{com}})$, we fit a separate property sub-map $F_{\text{prop}}^c$ on $\mathcal{D}_c = \{(\boldsymbol{v}^{(d)}, (t_n^{(d)}, y_n^{(d)})_{n=1}^{N_d}) \mid F_{\text{com}}(\boldsymbol{v}^{(d)}) = c\}$. The property map is defined by weights $\boldsymbol{W} \in \mathbb{R}^{L \times K \times B}$ and biases $\boldsymbol{\beta} \in \mathbb{R}^L$, where $B$ is the number of basis functions describing each of the shape functions of a GAM, and $L$ is the number of properties. To simplify the presentation, we assume that all variables are numerical.

**Raw Shape Functions** Property functions need to satisfy many constraints to make sure that the properties predicted by them actually describe a possible trajectory. For instance, if we have a composition $(s_{-+h})$, we need to make sure that the derivative at $t_0$ is negative. Or $x(t_0) < x(t_1)$ if the first motif is increasing. Our goal is to satisfy as many of these constraints as possible *by construction*. Because of that, we introduce a concept of *raw shape functions*. Given $L$ properties, we have $K \times L$ raw shape functions. Each of them is represented as a linear combination of $B$ basis

functions (e.g., B-Spline basis functions (de Boor, 1968)). We define the raw shape function of the $k^{\text{th}}$ feature for the $l^{\text{th}}$ property as $\bar{g}_k^{(l)} = \sum_{b=1}^{B} W_{l,k,b} \phi_{b,k}$, where $\phi_{b,k}$ is a $b^{\text{th}}$ basis function for the $k^{\text{th}}$ feature.

**From Raw Shape Functions to Properties** We can now get the properties from raw shape functions by appropriate transformations. For instance, if we need to have a positive property function (which is equivalent to having all shape functions and bias positive as we show in Appendix B), it is enough to apply a softplus function, denoted soft$_+$ and defined as $\text{soft}_+(z) = \log(1 + e^z)$, to each of the raw shape functions. The resulting GAM for the $l^{\text{th}}$ property function would be $\sum_{k=1}^{K} \text{soft}_+(\bar{g}_k^{(l)}(v_k)) + \text{soft}_+(\beta^{(l)})$. We discuss other kinds of transformations in Appendix C.

**Optimization** Before optimization, we precompute $\phi_{k,b}(v_k)$ for all $k \in [K]$ and $b \in [B]$. This allows us to efficiently calculate the shape functions as simple matrix products. Then the model is trained to find $\boldsymbol{W}$ and $\boldsymbol{\beta}$ that minimize

$$\frac{1}{D} \sum_{d=1}^{D} \frac{1}{N_d} \sum_{n=1}^{N_d} \left( \hat{x}(t_n^{(d)}) - y_n^{(d)} \right)^2 \quad (2)$$

where

$$\hat{x} = F_{\text{traj}} \left( F_{\text{com}}(\boldsymbol{v}^{(d)}), F_{\text{prop}}^c(\boldsymbol{v}^{(d)}; \boldsymbol{W}, \boldsymbol{\beta}) \right). \quad (3)$$

We optimize it using the L-BFGS algorithm (Liu & Nocedal, 1989).

## 5. Case Study: Novel Pharmacokinetic Model for Tacrolimus

In this study, we demonstrate the main advantages of our proposed approach while developing a novel population pharmacokinetic (PopPK) model for Tacrolimus.[2] That includes incorporating semantic inductive biases (Section 5.2), understanding the dynamics (Section 5.3), verifying the properties (Section 5.4), and revising the model to guarantee no unexpected behavior (Section 5.5). For that task, we use a real Tacrolimus dataset (Woillard et al., 2011) containing patient covariates, the dose of the drug, and drug plasma concentration measured over time (for more details, see Appendix D.1). We not only obtain a model performing better than the discovered ODEs and the expert-found PopPK model from the literature, but we can certify that our model is biologically plausible.

---

[2]Implementation of EPISODE is available at https://github.com/krzysztof-kacprzyk/EPISODE.

### 5.1. Tacrolimus and PopPK Models

Tacrolimus is an immunosuppressive drug primarily used to prevent organ rejection after kidney transplantation. PopPK models predict the drug concentration over time for an individual with particular characteristics. They are crucial for understanding the variability in drug concentrations within a patient population. This information is essential for optimizing drug dosing to improve drug safety and efficacy. Although black-box models have been used for this task, concerns remain about their lack of transparency and guarantees of appropriate behavior under every condition. ODE discovery can potentially deliver a transparent model, but getting insights, verifying the properties, and refining the model is challenging, as we illustrate below.

### 5.2. Prior Knowledge

EPISODE allows us to incorporate *semantic inductive biases*, i.e., prior knowledge we have about the behavior of the dynamical system. This is in contrast to *syntactic inductive biases* (i.e., information about the structural form of the equations) available in ODE discovery algorithms.

PopPK models describe the drug plasma concentration after the drug is administered as a function of time. Thus we expect that the concentration should decay to 0 as $t \rightarrow \infty$ (assuming no additional doses). We incorporate this information directly in our model by choosing the composition library $\mathcal{C}'$ to only contain compositions where the last motif is $s_{-+h}$ (decreasing, convex, and approaching a horizontal asymptote). As we also do not expect many trend changes, we constrain our model to the following four compositions: $(s_{-+h})$, $(s_{--b}, s_{-+h})$, $(s_{+-b}, s_{--b}, s_{-+h})$, $(s_{++b}, s_{+-b}, s_{--b}, s_{-+h})$.

Designing syntactic inductive biases is more challenging because, in many complex settings, we often do not have much knowledge about the structural form of the equation describing the phenomenon. In a PDS like this one, it is far from obvious whether the term $v_{\text{weight}}x(t)$ or $v_{\text{creat}}^2$ should appear in the final equation.

### 5.3. Understanding Dynamics

To understand the dynamics of our dynamical system we need to access the semantic representation of our model. Fortunately, this is readily available in the semantic predictor $F_{\text{sem}}$. No additional mathematical analysis is necessary. We can see that the composition map is very simple as it just predicts the composition $(s_{++b}, s_{+-b}, s_{--b}, s_{-+h})$ for all samples (see Figure 5). We can now be sure that, no matter for which patient we use our model, and what drug dose we choose, the trajectory is always going to have a correct shape. This, however, is not the only thing we need to verify about our model. We discuss it further in Section 5.4.

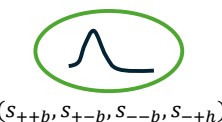

$$(s_{++b}, s_{+-b}, s_{--b}, s_{-+h})$$

*Figure 5.* Composition map found by our algorithm trained on the Tacrolimus dataset.

The equation found by SINDy constrained to 5 terms (SINDy-5) is the following.

$$\dot{x}(t) = 0.321v_{\text{HT}} - 21.543t - 6.42x(t)t$$
$$- 0.014v_{\text{HT}}v_{\text{HB}} + 19.065t^2 \tag{4}$$

Even though the equation is compact, it is not straightforward to understand the system dynamics without a careful mathematical analysis. Even answering a simple question of whether the concentration will increase at first cannot be answered directly. Comprehending what the rest of the trajectory looks like is even more challenging and the problem compounds for longer expressions.

### 5.4. Verification

Expert PopPK models have a structural form inspired by biological processes (e.g., movement of the drug between different organs, natural decay, or excretion). This usually guarantees that the models are *biologically plausible*. This, however, is not guaranteed in data-driven models. For the purpose of this section, we are just going to focus on two conditions that a biologically plausible PopPK model needs to adhere to.

- The concentration should initially increase after the drug is administered and then fall as the drug decays.
- The concentration should decay to 0.

With our approach, verifying these two conditions is straightforward. The composition map of the found model is very simple and assigns the composition $(s_{++b}, s_{+-b}, s_{--b}, s_{-+h})$ to all samples. This is precisely the shape we want. The function first increases, then decreases, and approaches a horizontal asymptote. We can also verify whether it decays to 0. We can look at the GAM that predicts the horizontal asymptote (all shape functions can be seen in Figure 6 in Appendix A.1). We can see that our model does not yet satisfy this condition. We will rectify this in the next subsection.

Verifying these two conditions for a closed-form ODE is potentially possible, but it may be time-consuming. Let us consider Equation (4) found by SINDy-5. Assume that the two conditions above hold. Then for some small $x_1$ there exists $t'$ large enough such that we have $\dot{x}(t) \leq 0$ and $0 \leq x(t) < x_1$ for all $t > t'$. Then for $t > t'$, $\dot{x}(t) > 0.321v_{\text{HT}} - 21.543t - 6.42x_1t - 0.014v_{\text{HT}}v_{\text{HB}} + 19.065t^2 \rightarrow$

$+\infty$ as $t \to +\infty$. So, there has to be $t > t'$ such that $\dot{x}(t) > 0$, which contradicts our conditions.

### 5.5. Editing

Editing the model is useful for both improving performance and satisfying some domain-specific criteria. In the previous section, we established that the horizontal asymptote of our model is not what it should be. We can just take one of the property maps and set it to 0. However, this may deteriorate performance, as it was not enforced during training. Instead, we can first encourage the asymptote to be close to 0 and only then set it to 0 exactly. As the semantic predictor directly predicts the semantic representation of the trajectory, we can put soft constraints on any aspect of the semantic representations, such as the coordinates of the transition points or the properties of the unbounded motif. In this case, we refit our model after adding a penalty $h^2$, where $h$ is the value of the horizontal asymptote. After training, we can see that the bias term (the expected value) is $-0.0019$, which is much closer to what we wanted (all shape functions can be seen in Figure 7 in Appendix A.1). Now, we can safely set the whole property map to 0 without a significant drop in performance (see Figure 8). Finally, we implemented a method to prune shape functions whose contribution (after being centered) is smaller than $1e-5$. Part of the model is visualized in Figure 1, and the property functions for the maximum and the time it is achieved can be seen in Figures 9 and 10 in Appendix A.2.

### 5.6. Performance

The performance of different methods, with different constraints, can be seen in Table 1. We use the following error metric (lower is better).

$$\frac{1}{D}\sum_{d=1}^{D}\sqrt{\frac{1}{N_d}\sum_{n=1}^{N_d}\left(\frac{1}{M}\left\|\boldsymbol{F}(\boldsymbol{v}^{(d)})\left(t_n^{(d)}\right)-\boldsymbol{y}_n^{(d)}\right\|_2^2\right)} \quad (5)$$

Here, $\mathbf{F}$ is the predictive model (e.g., EPISODE), $D$ is the number of samples, $M$ is the dimensionality of the system, $N_d$ is the number of measurements of the $d^{\text{th}}$ sample $(\boldsymbol{v}^{(d)},(t_n^{(d)},\boldsymbol{y}_n^{(d)})_{n=1}^{N_d})$. We choose this metric because for $M=1$ it reduces to the standard mean RMSE, i.e.,

$$\frac{1}{D}\sum_{d=1}^{D}\sqrt{\frac{1}{N_d}\sum_{n=1}^{N_d}\left(F(\boldsymbol{v}^{(d)})\left(t_n^{(d)}\right)-y_n^{(d)}\right)^2} \quad (6)$$

(used in Semantic ODE paper), and we normalize by $M$ so that it is easier to compare results between systems of different dimensionalities. The error is calculated on a held-out dataset of samples, whose trajectories are observed at the same time points as the ones in the training dataset. We observe that our method performs better than the found

*Table 1.* Results of fitting different models on the real Tacrolimus dataset. Performance on a test dataset. Standard deviation of the performance over the samples in brackets. Error measured using Equation (5). Lower is better.

| Black box models | | Closed-form expressions | |
|---|---|---|---|
| NeuralODE | $0.263_{(.111)}$ | SINDy-5 | $0.337_{(.144)}$ |
| ANODE | $0.268_{(.113)}$ | SINDy-20 | $0.331_{(.131)}$ |
| LatentODE | $0.281_{(.100)}$ | | |
| **Direct semantic modeling** | | WSINDy-5 | $0.348_{(.182)}$ |
| EPISODE | $0.255_{(.113)}$ | WSINDy-20 | $0.526_{(.531)}$ |
| EPISODE-verified | $0.257_{(.109)}$ | Expert PopPK | $0.351_{(.177)}$ |

ODEs and the expert model from the literature. It is also slightly better than unverifiable black-box approaches.

## 6. Flexibility

Although the most crucial aspects of our method are its transparency, verifiability, and editability, it is instructive whether it can also deliver a good performance in practice.

**Baselines** In this section, we compare our method against two ODE discovery approaches, **SINDy** (Brunton et al., 2016b) and **WSINDy** (Reinbold et al., 2020; Messenger & Bortz, 2021a) (as implemented in PySINDy (de Silva et al., 2020), in two different variants: up to 5 terms (compact) and up to 20 terms (not compact). Smaller equations are easier to interpret but may not be as flexible as long expressions. We also compare our approach to three black box approaches: **Neural ODE** (Chen et al., 2018), **ANODE** (Dupont et al., 2019), and **Latent ODE** (Rubanova et al., 2019) (non-probabilistic variant).

**EPISODE** For EPISODE, we choose a generous set of compositions without much prior knowledge. We consider all compositions up to length 4 (or length 8 for the Bike sharing dataset) except for compositions where two consecutive transition points are inflection points. This ends up being, respectively, 26 and 34 compositions. To demonstrate the power of how easily we can incorporate prior knowledge into our model, we also compare against our method with inductive bias about the possible compositions of the trajectory. We call this variant EPISODE*. The exact composition libraries used for each dataset are shown in Appendix D.3. Although the training process consists of two steps (training the composition map and then the property maps), the whole pipeline is end-to-end, and there is no manual intervention in experiments performed in this section.

**Datasets** We test the methods on a variety of datasets. To show that we can model traditional multi-dimensional

*Table 2.* Comparison between different methods tested on several synthetic and real datasets. † means that the method has not converged to a solution in at least one trial. The table shows means from 5 random dataset splits and seeds. Standard deviations in brackets. Error measured using Equation (5). Lower is better.

| | SIR | | HIV | | PK | | Tumor | | Tacrolimus | Bike sharing |
|---|---|---|---|---|---|---|---|---|---|---|
| | $\sigma = 0.01$ | $\sigma = 0.1$ | $\sigma = 0.01$ | $\sigma = 0.1$ | $\sigma = 0.01$ | $\sigma = 0.1$ | $\sigma = 0.01$ | $\sigma = 0.1$ | real | real |
| **Black box models** | | | | | | | | | | |
| NeuralODE | $0.021_{(.006)}$ | $0.102_{(.001)}$ | $0.188_{(.009)}$ | $0.214_{(.006)}$ | $0.156_{(.006)}$ | $0.182_{(.006)}$ | $0.300_{(.063)}$ | $0.334_{(.048)}$ | $0.267_{(.034)}$ | $0.178_{(.013)}$ |
| ANODE | $0.017_{(.003)}$ | $0.101_{(.001)}$ | $0.188_{(.008)}$ | $0.214_{(.005)}$ | $0.147_{(.005)}$ | $0.184_{(.009)}$ | $0.317_{(.049)}$ | $0.335_{(.049)}$ | $0.250_{(.011)}$ | $0.154_{(.011)}$ |
| LatentODE | $0.016_{(.002)}$ | $0.101_{(.001)}$ | $0.183_{(.006)}$ | $0.213_{(.006)}$ | $0.145_{(.004)}$ | $0.172_{(.003)}$ | $0.310_{(.051)}$ | $0.338_{(.052)}$ | $0.260_{(.013)}$ | $0.149_{(.006)}$ |
| **Closed-form expressions** | | | | | | | | | | |
| SINDy-5 | $0.014_{(.001)}$ | $0.135_{(.007)}$ | $0.175_{(.070)}$ | $0.284_{(.006)}$ | $0.211_{(.013)}$ | $0.249_{(.010)}$ | $0.095_{(.018)}$ | $0.151_{(.012)}$ | $0.316_{(.032)}$ | $0.267_{(.010)}$ |
| SINDy-20 | $0.015_{(.001)}$ | $0.142_{(.003)}$ | † | † | $0.180_{(.007)}$ | † | $0.062_{(.015)}$ | $0.138_{(.013)}$ | $0.346_{(.040)}$ | $0.250_{(.012)}$ |
| WSINDy-5 | $0.026_{(.025)}$ | $0.120_{(.006)}$ | $0.246_{(.082)}$ | † | $0.279_{(.009)}$ | $0.309_{(.016)}$ | $0.094_{(.024)}$ | $0.145_{(.017)}$ | $0.415_{(.190)}$ | † |
| WSINDy-20 | $0.016_{(.001)}$ | $0.144_{(.010)}$ | † | † | † | † | $0.056_{(.014)}$ | $0.129_{(.008)}$ | $0.816_{(.670)}$ | $0.310_{(.069)}$ |
| **Direct semantic modeling** | | | | | | | | | | |
| EPISODE | $0.026_{(.004)}$ | $0.103_{(.001)}$ | $0.088_{(.006)}$ | $0.143_{(.007)}$ | $0.088_{(.008)}$ | $0.132_{(.013)}$ | $0.144_{(.032)}$ | $0.290_{(.147)}$ | $0.256_{(.007)}$ | $0.192_{(.013)}$ |
| EPISODE* | $0.012_{(.000)}$ | $0.100_{(.001)}$ | $0.089_{(.006)}$ | $0.142_{(.007)}$ | $0.095_{(.039)}$ | $0.146_{(.037)}$ | $0.098_{(.014)}$ | $0.165_{(.016)}$ | $0.256_{(.007)}$ | $0.195_{(.029)}$ |

systems, we test the models on the **SIR** epidemiological model and **HIV** model (Hill et al., 2018). We also test on synthetic tumor growth (**Tumor**) and pharmacokinetic system (**PK**) as well as on real **Tacrolimus** and **Bicycle sharing** datasets. For synthetic datasets, we consider two different noise settings: low ($\sigma = 0.01$) and high ($\sigma = 0.1$). Details about all datasets can be found in Appendix D.1.

**Results**   The results can be seen in Table 2. The error metric used is the same as in Section 5.6, i.e., Equation (5). We show sample trajectories predicted by EPISODE on those datasets in Figure 11 in Appendix A.3. We can see that EPISODE or EPISODE* outperforms other approaches on most datasets. Performance on the SIR datasets shows that it can compete even on problems where ODE-based methods have an advantage (a system of ODEs describing the dynamics in the search space of SINDy). It is promising that EPISODE can outperform even black-box approaches in certain settings. This can be explained by some of the inductive biases playing the role of regularizers (e.g., enforcing smoothness or a small number of inflection points).

## 7. Discussion

**Applications**   EPISODE revolutionizes the modeling of personalized dynamical systems by making them transparent, editable, and intuitive. Unlike traditional ODE discovery, EPISODE skips complex analysis, directly predicting system behavior for rapid validation and real-time adjustments. This empowers experts to refine models dynamically, ensuring accuracy, reliability, and trust—crucial for high-stakes applications. By bridging machine learning with human intuition, EPISODE makes scientific modeling not just powerful, but truly usable.

**Limitations and Open Challenges**   Although EPISODE resolves one of the biggest shortcomings of Semantic ODEs,

making DSM applicable in settings with multi-dimensional inputs, some of the original issues remain. That includes restriction to finite compositions which prevents modeling systems with oscillatory or periodic behavior. Fitting the composition map remains the most time-consuming part of the training due to the necessity of matching every composition to each sample. Future research should focus on finding more efficient methods of fitting the composition map.

## Acknowledgements

This work was supported by Roche and AstraZeneca. We would like to thank Harry Amad, Benjamin Lapostolle, Victor Baillet, and anonymous reviewers for their useful comments and feedback on earlier versions of this work.

## Impact Statement

This paper presents a new method to improve understanding of a dynamical system's behavior through direct semantic modeling in personalized dynamical systems. Enhancing transparency in machine learning models is essential for tasks such as debugging, meeting domain-specific regulations, and mitigating potential biases. However, if not used correctly, these techniques may create a misleading sense of security regarding model outputs or be used merely for nominal compliance with regulations. Given that our method is relevant to critical fields like medicine and pharmacology, it is crucial to perform a comprehensive evaluation before implementing the model in these sensitive areas. Such assessments should confirm that the model's performance aligns with ethical standards and does not result in decisions that could harm individuals' health and well-being.

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

## Table of supplementary materials

## A. Additional Results

### A.1. Horizontal Asymptote

Figure 6 shows a GAM describing the horizontal asymptote of the composition $(s_{++b}, s_{+-b}, s_{--b}, s_{-+h})$ of the original EPISODE trained on the Tacrolimus dataset. Figure 7 shows how this property function changes as we add a penalty term encouraging the asymptote to be close to 0. Finally, Figure 8 shows the property function after it is set to 0 to make the model biologically plausible.

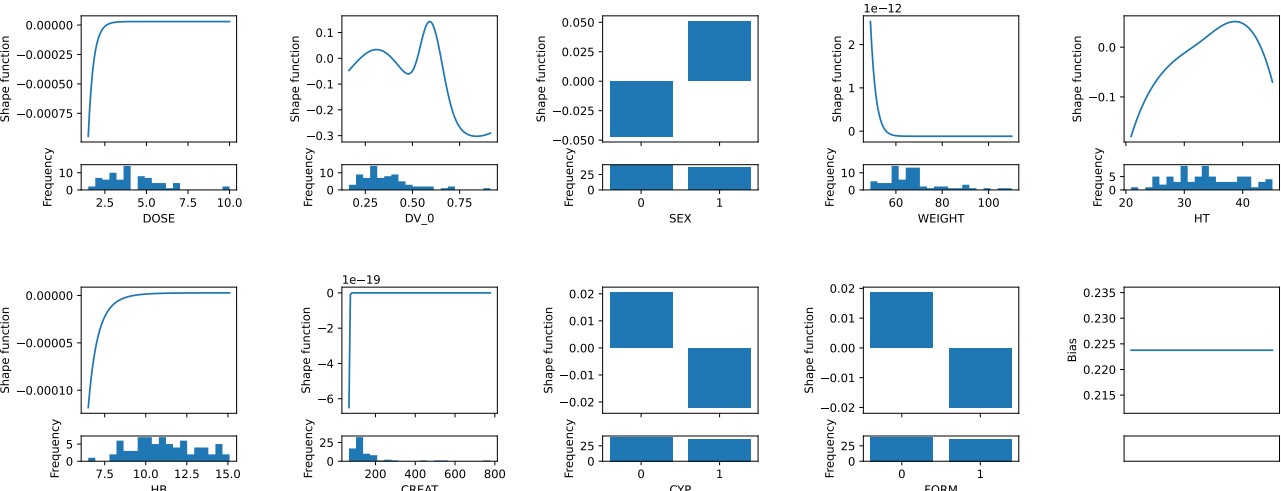

*Figure 6.* The original GAM describing the horizontal asymptote of the composition $(s_{++b}, s_{+-b}, s_{--b}, s_{-+h})$ trained on the Tacrolimus dataset. To make it biologically plausible, it should have an asymptote equal to 0. We describe how to fix it in Section 5.5

### A.2. Maximum of the Trajectory

Figure 9 shows the property function describing the maximum of the trajectory for the model trained on the Tacrolimus dataset. Figure 10 describes the time at which it is achieved. Both GAMs have been pruned to remove shape functions that contribute less than $1e - 5$.

### A.3. Examples Trajectories

Figure 11 shows sample trajectories predicted on three datasets used in Section 6.

## B. Theory

To realize direct semantic modeling, we need the following three ingredients.

1. Definition of the semantic representation of an $M$-dimensional trajectory $\boldsymbol{x} \in C^2(\mathcal{T})$.

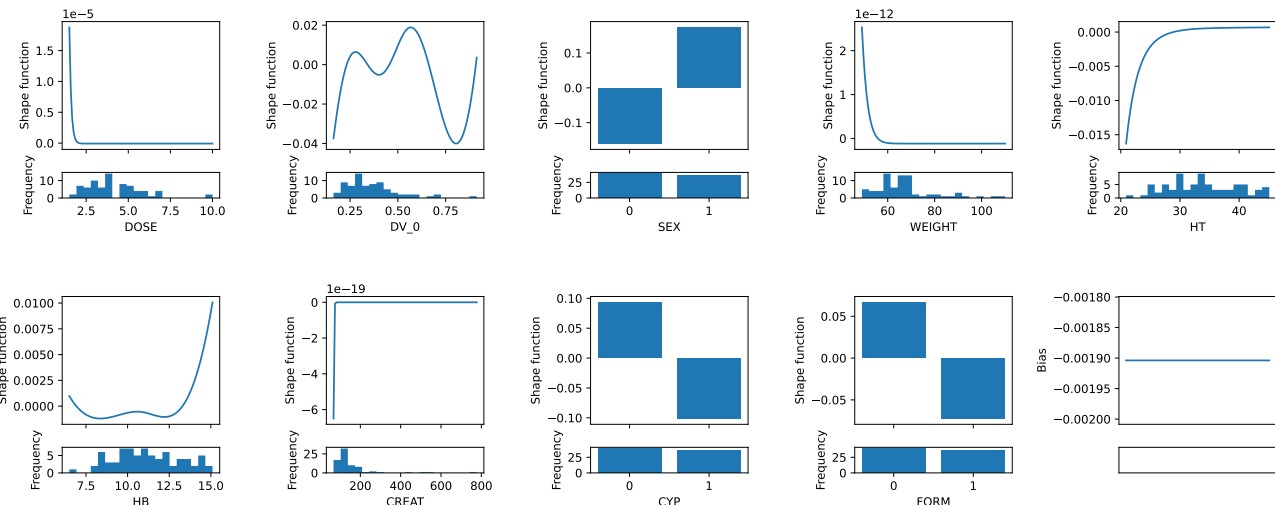

*Figure 7.* GAM describing the horizontal asymptote of the composition $(s_{++b}, s_{+-b}, s_{--b}, s_{-+h})$ trained on the Tacrolimus dataset after adding a penalty term encouraging the horizontal asymptote to be close to 0. We can now safely set it to 0 without worrying too much about lost performance.

2. Trajectory predictor $\boldsymbol{F}_{\text{traj}}$ that output a trajectory $\boldsymbol{x}$ given a semantic representation of this trajectory.
3. Semantic predictor $F_{\text{sem}}$ that outputs a semantic representation of $\boldsymbol{x}$ given a vector of static features $\boldsymbol{v} \in \mathcal{V}$.

More formally, the semantic representation of an $M$-dimensional trajectory $\boldsymbol{x}$ is defined as follows.

**Definition B.1** (Semantic representation of $\boldsymbol{x}$). Let $\boldsymbol{x}$ be an $M$-dimensional trajectory. The semantic representation of $\boldsymbol{x}$ is a pair $(\boldsymbol{c_x}, \boldsymbol{p_x})$, where $\boldsymbol{c_x}$ and $\boldsymbol{p_x}$ are vectors of entries defined as $(\boldsymbol{c_x})_m = c_{x_m}$ and $(\boldsymbol{p_x})_m = p_{x_m}$.

We can similarly define $\boldsymbol{F}_{\text{traj}}$.

**Definition B.2** (Trajectory predictor). The $m^{\text{th}}$ coordinate of the trajectory $\boldsymbol{x}$ predicted by the trajectory predictor $\boldsymbol{F}_{\text{traj}}$ is defined as $x_m = F_{\text{traj}}^{(m)}(\boldsymbol{c}, \boldsymbol{p}) = F_{\text{traj}}(c_m, p_m)$.

Of course, such trajectory predictor is consistent, i.e., the semantic representation of the trajectory predicted by the trajectory predictor is equal to the original semantic representation.

**Proposition B.3** (Consistency of trajectory predictor). *Let $(\boldsymbol{c}, \boldsymbol{p})$ be a semantic representation of some trajectory. Let $\boldsymbol{x} = \boldsymbol{F}_{traj}(\boldsymbol{c}, \boldsymbol{p})$. Then $(\boldsymbol{c_x}, \boldsymbol{p_x}) = (\boldsymbol{c}, \boldsymbol{p})$.*

*Proof.* From Definition B.2, $x_m = F_{\text{traj}}^{(m)}(\boldsymbol{c}, \boldsymbol{p}) = F_{\text{traj}}(c_m, p_m)$. From $x_m = F_{\text{traj}}(c_m, p_m)$ and by the original definition of $F_{\text{traj}}$, we get that $(c_{x_m}, p_{x_m}) = (c_m, p_m)$. From Definition B.1, $c_{x_m} = (\boldsymbol{c_x})_m$ and $p_{x_m} = (\boldsymbol{p_x})_m$. Thus $(\boldsymbol{c_x})_m = c_m$ and $(\boldsymbol{p_x})_m = p_m$ for each $m$. Therefore $(\boldsymbol{c_x}, \boldsymbol{p_x}) = (\boldsymbol{c}, \boldsymbol{p})$. $\square$

**Positive GAMs**

**Proposition B.4.** *Let $h(\boldsymbol{v}) = \sum_{k=1}^{K} g_k(v_k) + g_0$ be a GAM. Let us assume that $h$ is positive and continuous for all possible inputs from a hyperbox input domain $\mathcal{V} = [a_1, b_1] \times \ldots \times [a_K, b_K]$. Then $h$ can be represented using positive shape functions and a positive bias term.*

*Proof.* Let $\gamma_k = \min_{v \in [a_k, b_k]} g_k(v)$ for each $k \in [K]$. If $h$ is positive and continuous for all possible inputs from a hyperbox then there exists $\epsilon > 0$ such that $h(\boldsymbol{v}) > \epsilon$ for all $\boldsymbol{v} \in \mathcal{V}$. Let us define new shape functions as

$$\bar{g}_k(v_k) = g_k(v_k) - \gamma_k + \epsilon/(K+1) \tag{7}$$

Then the new bias term needs to be equal to

$$\bar{g}_0 = g_0 - \sum_{k=1}^{K} (\bar{g}_k(v_k) - g_k(v_k)) \tag{8}$$

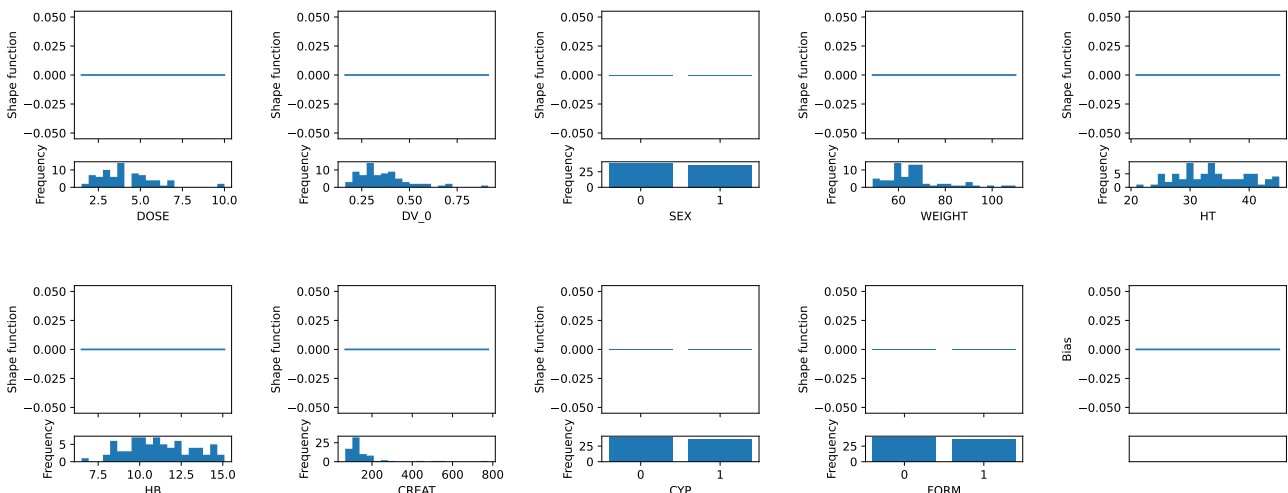

*Figure 8.* GAM describing the horizontal asymptote of the composition $(s_{++b}, s_{+-b}, s_{--b}, s_{-+h})$ trained on the Tacrolimus dataset after setting it to 0.

Thus $\bar{g}_0 = g_0 - \sum_{k=1}^{K}(-\gamma_k + \epsilon/(K+1))$. This means

$$\bar{g}_0 = g_0 + \sum_{k=1}^{K} \gamma_k - \epsilon \times \frac{K}{K+1} \tag{9}$$

As $\mathcal{V}$ is a hyperbox then it must hold that $\sum_{k=1}^{K} \gamma_k + g_0 > \epsilon$. That means that

$$\bar{g}_0 > \epsilon \times \left(1 - \frac{K}{K+1}\right) = \epsilon/(K+1) > 0 \tag{10}$$

Thus all shape functions and the bias term are positive. $\square$

## C. Implementation

### C.1. Block diagrams

Figure 12 shows a block diagram of how a single semantic predictor works. Figure 13 shows the overall training procedure of a single semantic predictor.

### C.2. Code

The code of the implementation and the experiments is available at https://github.com/krzysztof-kacprzyk/EPISODE and https://github.com/vanderschaarlab/EPISODE.

### C.3. Composition map

To find a decision tree that minimizes such an objective, we perform a greedy search where, at each node, we evaluate all possible splits for all possible features and choose the one that gives the smallest error. Consider a leaf of a partially fitted tree and a subset of $[D]$ denoted $\mathcal{I}$ such that all $\boldsymbol{v}^{(d)}$ for $d \in \mathcal{I}$ satisfy all conditions from root to this leaf. We define the current error at the leaf to be $E(\mathcal{I}) = \min_{c \in \mathcal{C}} \frac{1}{|\mathcal{I}|} \sum_{d \in \mathcal{I}} e^{(d)}[c]$. The goal is to find a feature split that divides $\mathcal{I}$ into $\mathcal{I}_1$ and $\mathcal{I}_2$ such that $\frac{|\mathcal{I}_1|}{|\mathcal{I}|} E(\mathcal{I}_1) + \frac{|\mathcal{I}_2|}{|\mathcal{I}|} E(\mathcal{I}_2) < E(\mathcal{I})$. In practice, we put a few additional constraints to prevent a composition map that is unnecessarily more complex without a significant improvement in performance. For instance, we set a maximum depth of the tree or a minimum relative improvement for a split to occur. We also penalize longer compositions. We set a minimum number of samples in a leaf so that we have enough data to train a property sub-map for the composition predicted by that leaf. Finally, we train trees for different maximum depths and choose the one with the lowest validation loss to prevent overfitting.

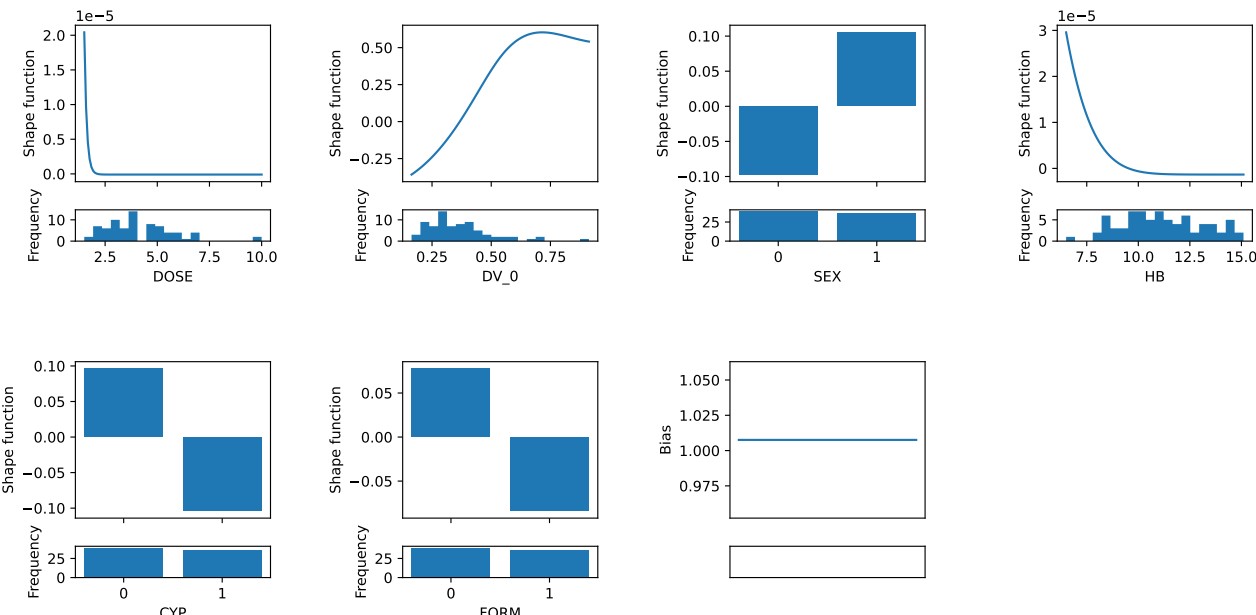

*Figure 9.* GAM describing the maximum value of the composition $(s_{++b}, s_{+-b}, s_{--b}, s_{-+h})$ trained on the Tacrolimus dataset. This property function has been pruned to remove shape functions that contribute less than $1e-5$

## C.4. Property map

Property functions need to satisfy many constraints to make sure that the properties predicted by them actually describe a possible trajectory. For instance, property functions describing $x(t_0)$ and $x(t_1)$ need to satisfy $x(t_0) < x(t_1)$ for all possible $v$ if the first motif is increasing. To achieve that, we design the property maps so that they satisfy most of these constraints by construction. For instance, instead of fitting the property function for $x(t_1)$ directly, we fit a GAM for the *difference* between $x(t_0)$ and $x(t_1)$. As long as we make this difference positive (or negative if the first motif is decreasing), we can add this GAM to the one describing $x(t_0)$ and get a valid property function. How can we get a GAM that is always positive? We observe that if a GAM is positive for all possible inputs (from a hyperbox input domain), then it is always possible to represent it using only positive shape functions and a positive bias (proof in Appendix B). Thus, we can limit ourselves to only such shape functions. Ultimately the property function for $x(t_1)$ evaluated at $v$ is given by

$$\sum_{k=1}^{K} \text{soft}_+ \left( \bar{g}_k^{(1)}(v_k) \right) + \text{soft}_+(\beta_1) + x(t_0) \tag{11}$$

This not only guarantees that $x(t_1) > x(t_0)$ but, crucially, as $x(t_0)$ is also described by (unconstrained) GAM, $x(t_1)$ is still a GAM where the $k^{\text{th}}$ shape function is given by

$$\text{soft}_+ \left( \bar{g}_k^{(1)}(v_k) \right) + \bar{g}_k^{(0)}(v_k) \tag{12}$$

and the bias term is equal to $\text{soft}_+(\beta_1) + \beta_0$.

**Works Well With Categorical Features**   Because GAMs work very naturally with categorical features ($g_k$ just assigns a different value to each category), this makes our model much more appropriate than ODEs in settings where categorical features are present. ODEs, by default, can only work with numerical variables, and one-hot encoding categorical features may result in complicated equations.

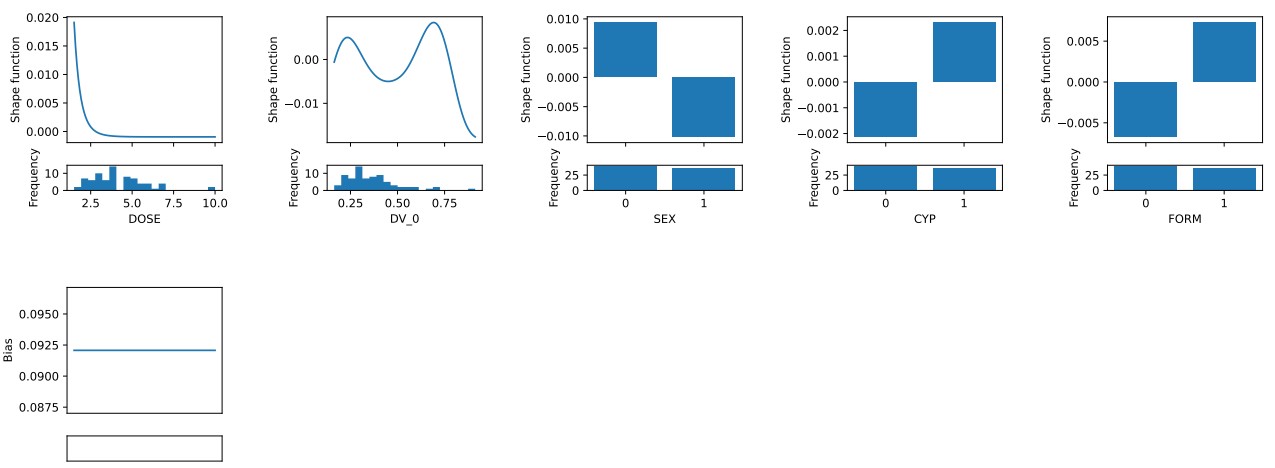

*Figure 10.* GAM describing the time at which the maximum is obtained of the composition $(s_{++b}, s_{+-b}, s_{--b}, s_{-+h})$ trained on the Tacrolimus dataset. This property function has been pruned to remove shape functions that contribute less than $1e-5$

## D. Experimental Details

### D.1. Datasets

**SIR**    The SIR dataset is a synthetic dataset based on an SIR epidemiological model (Kermack et al., 1997). It is a three-dimensional dynamical system governed by the following system of ODEs,

$$\frac{\mathrm{d}S}{\mathrm{d}t} = -\beta SI \tag{13}$$

$$\frac{\mathrm{d}I}{\mathrm{d}t} = \beta SI - \gamma I \tag{14}$$

$$\frac{\mathrm{d}R}{\mathrm{d}t} = \gamma I, \tag{15}$$

and initial conditions

$$S(0) = S_0 \tag{16}$$
$$I(0) = I_0 \tag{17}$$
$$R(0) = R_0. \tag{18}$$

We set $\beta = 0.3$ and $\gamma = 0.1$. We sample $S_0$ uniformly from $(0.6, 1.0)$, $I_0$ uniformly from $(0.01, 0.1$, $R_0$ uniformly from $(0.0, 1.0)$. We generate 500 trajectories, each measured at 20 equally spaced time points between 0 and 100. Finally, we add Gaussian noise to each measurement with $\sigma = 0.01$ and divide the time points by 100 to scale them to interval $(0, 1)$. The set of static features contains only the initial conditions, i.e., $\boldsymbol{v} = (S_0, I_0, R_0)$.

**PK**    The PK dataset is a synthetic dataset based on a pharmacokinetic model published by Woillard et al. (2011). It predicts the drug concentration over time based on patient covariates and the dose of the drug. It has a form of a system of ODEs and covariate models describing the parameters of the ODEs in terms of patients' covariates. It consists of the following system

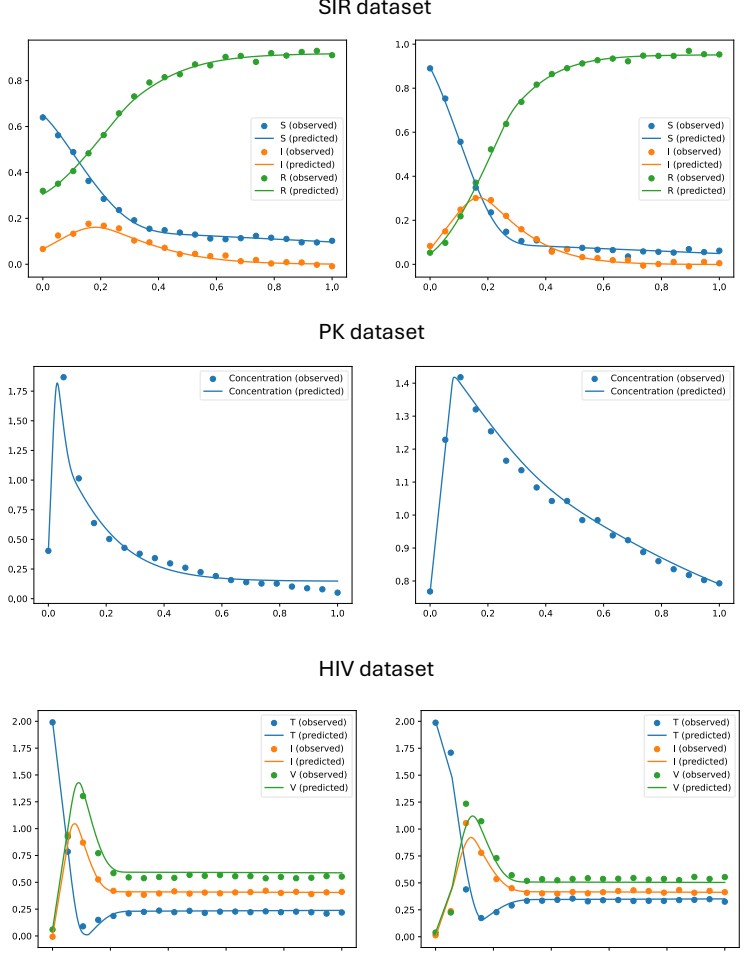

*Figure 11.* Sample trajectories predicted by EPISODE*. top row: SIR dataset ($\sigma = 0.01$), middle row: PK dataset ($\sigma = 0.01$), bottom row: HIV dataset ($\sigma = 0.01$).

of ODEs.

$$\frac{\mathrm{d}C_{\text{depot}}}{\mathrm{d}t} = -k_{\text{tr}}C_{\text{depot}} \tag{19}$$

$$\frac{\mathrm{d}C_{\text{trans1}}}{\mathrm{d}t} = k_{\text{tr}}C_{\text{depot}} - k_{\text{tr}}C_{\text{trans1}} \tag{20}$$

$$\frac{\mathrm{d}C_{\text{trans2}}}{\mathrm{d}t} = k_{\text{tr}}C_{\text{trans1}} - k_{\text{tr}}C_{\text{trans2}} \tag{21}$$

$$\frac{\mathrm{d}C_{\text{trans3}}}{\mathrm{d}t} = k_{\text{tr}}C_{\text{trans2}} - k_{\text{tr}}C_{\text{trans3}} \tag{22}$$

$$\frac{\mathrm{d}C_{\text{cent}}}{\mathrm{d}t} = k_{\text{tr}}C_{\text{trans3}} - ((CL + Q) * C_{\text{cent}}/V_1) + (Q * C_{\text{peri}}/V_2) \tag{23}$$

$$\frac{\mathrm{d}C_{\text{peri}}}{\mathrm{d}t} = (Q * C_{\text{cent}}/V_1) - (Q * C_{\text{peri}}/V_2) \tag{24}$$

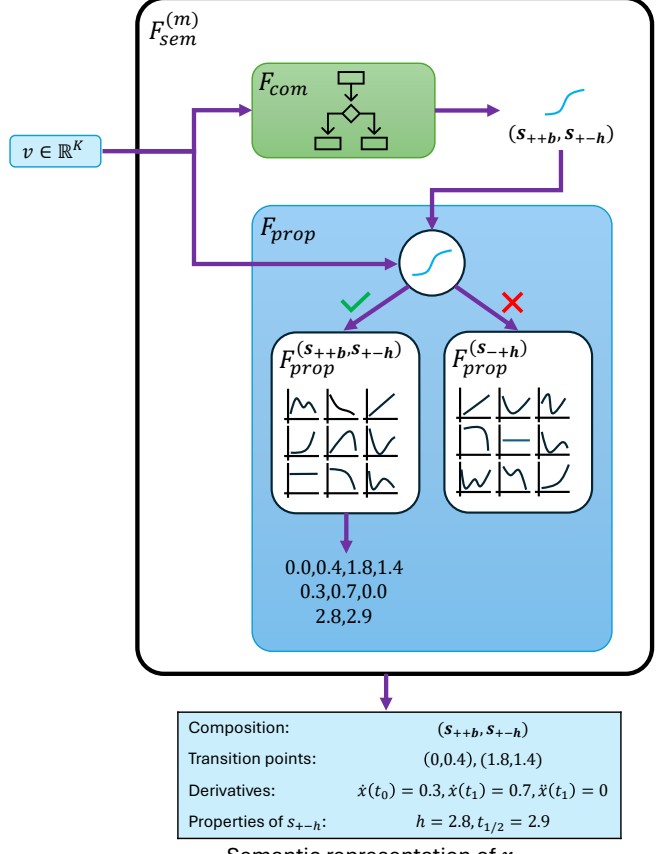

Composition: $(\mathbf{s}_{++b}, \mathbf{s}_{+-h})$
Transition points: $(0, 0.4)$, $(1.8, 1.4)$
Derivatives: $\dot{x}(t_0) = 0.3, \dot{x}(t_1) = 0.7, \ddot{x}(t_1) = 0$
Properties of $s_{+-h}$: $h = 2.8, t_{1/2} = 2.9$

Semantic representation of $x_m$

*Figure 12.* Block diagram depicting a single semantic predictor

The initial conditions are:

$$C_{\text{depot}}(0) = v_{\text{DOSE}} \tag{25}$$

$$C_{\text{trans1}}(0) = 0 \tag{26}$$

$$C_{\text{trans2}}(0) = 0 \tag{27}$$

$$C_{\text{trans3}}(0) = 0 \tag{28}$$

$$C_{\text{cent}}(0) = v_{\text{DV0}} \times \frac{V_1}{1000} \tag{29}$$

$$C_{\text{peri}}(0) = \frac{V_2}{V_1} \times C_{\text{cent}}(0). \tag{30}$$

The parameters are described as follows.

$$\text{CL} = \text{TVCL} \times \left(\frac{v_{\text{HT}}}{35}\right)^{\text{HTCL}} \times \text{CYPCL}^{v_{\text{CYP}}} \tag{31}$$

$$V_1 = \text{TVV1} \times \text{STV1}^{v_{\text{FORM}}} \tag{32}$$

$$Q = \text{TVQ} \tag{33}$$

$$V_2 = \text{TVV2} \tag{34}$$

$$k_{\text{tr}} = \text{TVKTR} \times \text{STKTR}^{v_{\text{FORM}}} \tag{35}$$

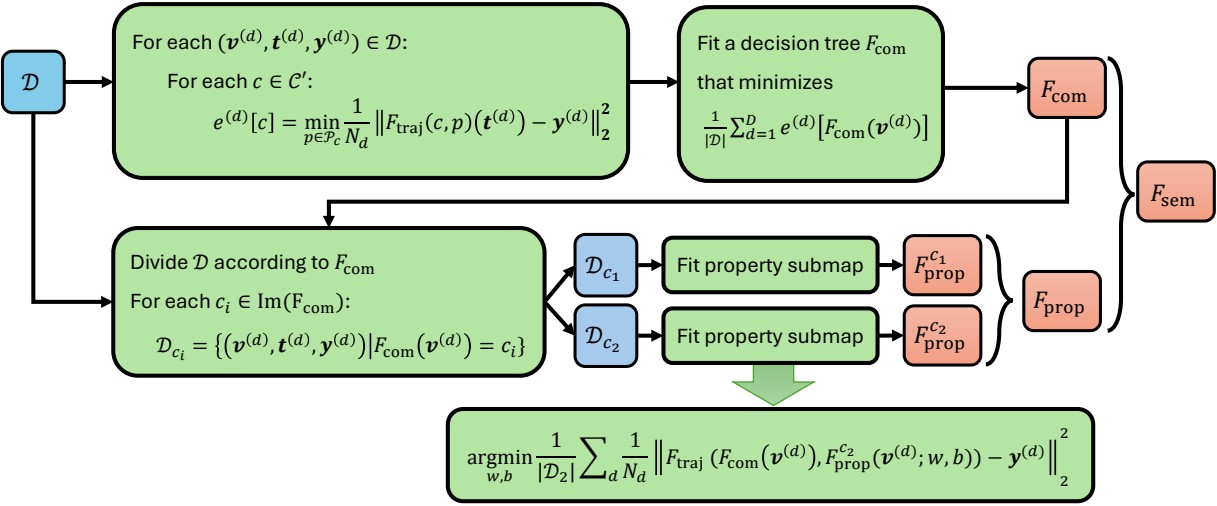

*Figure 13.* Block diagram depicting the general training procedure

The constants are defined as follows.

$$\text{TVCL} = 21.2 \tag{36}$$
$$\text{TVV1} = 486 \tag{37}$$
$$\text{TVQ} = 79 \tag{38}$$
$$\text{TVV2} = 271 \tag{39}$$
$$\text{TVKTR} = 3.34 \tag{40}$$
$$\text{HTCL} = -1.14 \tag{41}$$
$$\text{CYPCL} = 2.00 \tag{42}$$
$$\text{STKTR} = 1.53 \tag{43}$$
$$\text{STV1} = 0.29 \tag{44}$$
$$\tag{45}$$

We generate 500 patients by uniformly sampling the static features from the ranges described in Table 3. Then we solve the initial value problem to get all trajectories but return (observe) only $C_{\text{cent}}$. We then convert to proper units by multiplying by $\frac{1000}{V_1}$. Each trajectory is measured at 20 equally spaced time points between 0 and 24. Then we divide the time points by 24 and the trajectories by 20 to get them in a range $(0, 1)$. We also divide $v_{\text{DV0}}$ by 20 as it corresponds to the initial value. Finally, we add Gaussian noise with $\sigma = 0.01$.

**Tumor**   The Tumor dataset is a synthetic dataset used by Kacprzyk et al. (2024b). It is based on the tumor growth model proposed by Wilkerson et al. (2017). It is described by the following equation.

$$x(t) = \phi \exp(-dt) + (1 - \phi)\exp(gt) - 1 \tag{46}$$

Where $\phi, d, g$ are defined as follows.

$$
\begin{aligned}
g &= g_0 * (v_{\text{age}}/20.0)^{0.5} \\
d &= d_0 * v_{\text{dosage}}/v_{\text{weight}} \\
\phi &= 1/(1 + \exp(-v_{\text{dosage}} * \phi_0))
\end{aligned}
\tag{47}
$$

*Table 3.* Feature ranges in PK dataset

| Feature | Range |
|---------|-------|
| $v_{\text{DOSE}}$ | (1,10) |
| $v_{\text{DV0}}$ | (0,20) |
| $v_{\text{SEX}}$ | 0,1 |
| $v_{\text{WEIGHT}}$ | (45,110) |
| $v_{\text{HT}}$ | (20,47) |
| $v_{\text{HB}}$ | (6,16) |
| $v_{\text{CREAT}}$ | (60,830) |
| $v_{\text{CYP}}$ | 0,1 |
| $v_{\text{FORM}}$ | 0,1 |

And where $g_0, d_0, \phi_0$ are set to

$$
\begin{aligned}
g_0 &= 2.0 \\
d_0 &= 180 \\
\phi_0 &= 10.
\end{aligned}
\tag{48}
$$

We generate 500 samples by uniformly sampling the static features from the feature ranges described in Table 4. Each trajectory is measured at 20 equally spaced points on $(0, 1)$. Finally, we add Gaussian noise with $\sigma = 0.01$.

*Table 4.* Feature ranges in Tumor dataset

| Feature | Range |
|---------|-------|
| $v_{\text{age}}$ | (20,80) |
| $v_{\text{weight}}$ | (40,100) |
| $v_{\text{intitial-tumor-volume}}$ | (0.1,0.5) |
| $v_{\text{dose}}$ | (0.0,1.0) |

**Tacrolimus**   We take the drug concentration curves of two tacrolimus formulations obtained by (Woillard et al., 2011). The static features include sex, weight, hematocrit, hemoglobin, creatinine, dose, CYP3A5 genotype, and formulation. We also supplement it with the initial concentration of the drug in the blood. After preprocessing, we end up with 90 samples, each with 10 measurements on average. As not all trajectories are measured at exactly the same time points, we perform linear interpolation for each trajectory and sample it at the following time points: 0, 0.33, 0.67, 1, 1.5, 2, 3, 4, 6, 9, 12. Finally, we divide the time points by 12 and the trajectories by 20 to bring both of them into a $(0, 1)$ range. We also divide the initial condition by 20.

**Bike sharing**   We use a real bike-sharing dataset from the UCI repository (Fanaee-T, 2013). We perform preprocessing so that each sample corresponds to one day measured 24 times (every hour). As features $v_{\text{weathersit}}$, $v_{\text{temp}}$, $v_{\text{hum}}$, and $v_{\text{windspeed}}$ change throughout the day, we take their average to make them into static features. We also round $v_{\text{weathersit}}$ to keep it a categorical variable. We also add the value at $t = 0$ as an additional feature. At the end we have the following features: $v_{x_0}$, $v_{\text{season}}$, $v_{\text{month}}$, $v_{\text{workingday}}$, $v_{\text{weathersit}}$, $v_{\text{temp}}$, $v_{\text{hum}}$, $v_{\text{windspeed}}$. We divide the trajectories and $v_{x_0}$ by 500 to scale it and the time points by 23 to bring them to $(0, 1)$ range.

### D.2. Baselines

**SINDy**   We use SINDy (Brunton et al., 2016b) as implemented in PySINDy package (de Silva et al., 2020; Kaptanoglu et al., 2022). We pass the variable $t$ and the static features $\boldsymbol{v}$ as control variables. We use a library with all polynomial terms up to the second degree, including interaction between different variables. We use Mixed-Integer Optimized Sparse Regression (MIOSR) (Bertsimas & Gurnee, 2023) for optimization as it allows us to choose a sparsity level—the number of

terms in the equation. In our experiments, we consider two variants of SINDy, that we denote SINDy-5 and SINDy-20, enforcing the maximum number of terms to be, respectively, 5 and 20. We tune the parameters $\alpha$ of MIOSR that describes the strength of L2 penalty (between $1e-3$ and 1), and the derivative estimation algorithm using `Optuna` (Akiba et al., 2019) for 30 trials. We choose between the following techniques: finite difference, spline, trend filtered, and smoothed finite difference as available in PySINDy. The parameter ranges we consider for each of them are shown in Table 5.

*Table 5.* Hyperparameter ranges for each of the derivative estimation methods.

| Method | Hyperparameter ranges |
|---|---|
| finite difference | $k \in \{1, \ldots, 5\}$ |
| spline | $s \in (1e-3, 1)$ |
| trend filtered | order $\in \{0, 1, 2\}$, $\alpha \in (1e-4, 1)$ |
| smoothed finite difference | window length $\in \{1, \ldots, 5\}$ |

**WSINDy**   We use WSINDy (Reinbold et al., 2020; Messenger & Bortz, 2021a) based on (Reinbold et al., 2020) as implemented in PySINDy package (de Silva et al., 2020; Kaptanoglu et al., 2022). We pass the variable $t$ and the static features $v$ as control variables. We use a library with all polynomial terms up to the second degree, including interaction between different variables. We use Mixed-Integer Optimized Sparse Regression (MIOSR) (Bertsimas & Gurnee, 2023) for optimization as it allows us to choose a sparsity level—the number of terms in the equation. In our experiments, we consider two variants of WSINDy, that we denote WSINDy-5 and WSINDy-20, enforcing the maximum number of terms to be, respectively, 5 and 20. We tune the parameter $\alpha$ of MIOSR that describes the strength of the L2 penalty (between $1e-3$ and 1) using `Optuna` (Akiba et al., 2019) for 30 trials. We choose the parameter $K$ (the number of domain centers) to be 200.

**NeuralODE**   We implement a NeuralODE (Chen et al., 2018) model using `torchdiffeq` library (Chen, 2018). We parametrize the ODE as a neural network that takes $x$, $t$, and $v$ as inputs and outputs an $M$-dimensional vector. We then append a vector of zeros of size $K$ to account for the static features being constant (having derivative 0). The network consists of fully connected layers. The categorical variables are one-hot-encoded before training and the data is standardized. We set the batch size to 32 and train for 200 epochs using Adam optimizer (Kingma & Ba, 2017). We tune hyperparameters using `Optuna` (Akiba et al., 2019) for 20 trials. Ranges for the hyperparameters are shown in Table 6.

*Table 6.* Hyperparameter ranges used for tuning Neural ODE.

| Hyperparameter | Range |
|---|---|
| learning rate | (1e-5,1e-1) |
| number of layers | (1,3) |
| units in each layer (separately) | (16,128) |
| dropout rate | (0.0,0.5) |
| weight decay | (1e-6,1e-2) |
| activation function | ELU, Sigmoid |

**ANODE**   We implemented ANODE (Dupont et al., 2019) similarly to NeuralODE with the difference that we augment the $x$ by a few dimensions and evolve this extended dynamical system. We set the initial conditions for these additional dimensions to 0. We train and tune in the same way as NeuralODE with the only difference being that we also tune the number of additional dimensions. We consider a range between 1 and 10.

**LatentODE**   We implement a non-probabilistic variant of LatentODE (Rubanova et al., 2019). It is similar to NeuralODE with the difference being that the ODE is solved in the latent space, so we also learn an encoder and decoder. We tune hyperparameters using `Optuna` (Akiba et al., 2019) for 20 trials. Ranges for the hyperparameters are shown in Table 7.

### D.3. EPISODE

The composition libraries used in EPISODE* depend on the particular dataset and are as follows.

*Table 7.* Hyperparameter ranges used for tuning Latent ODE.

| Hyperparameter | Range |
|---|---|
| learning rate | (1e-5,1e-1) |
| number of layers in the encoder | (1,3) |
| units in each layer of the encoder (separately) | (16,128) |
| number of layers in decoder | (1,3) |
| units in each layer of decoder (separately) | (16,128) |
| number of layers in the derivative | (1,3) |
| units in each layer of derivative (separately) | (16,128) |
| latent dimension | (1,10) |
| dropout rate | (0.0,0.5) |
| weight decay | (1e-6,1e-2) |
| activation function | ELU, Sigmoid |

- SIR. $m = 1$: $\{(s_{--b}, s_{-+h})\}$, $m = 2$: $\{(s_{++b}, s_{+-b}, s_{--b}, s_{-+h})\}$, $m = 3$: $\{(s_{++b}, s_{+-h}\}$.

- PK. $m = 1$: $\{(s_{++b}, s_{+-b}, s_{--b}, s_{-+h})\}$.

- Tumor. $m = 1$: $\{(s_{++u}), (s_{-+b}, s_{++u}), (s_{-+h}), (s_{--b}, s_{-+h})\}$.

- Tacrolimus. $m = 1$: $\{(s_{++b}, s_{+-b}, s_{--b}, s_{-+h})\}$.

- Bike sharing. $m = 1$: $\{(s_{++b}, s_{+-b}, s_{--b}, s_{-+b}), (s_{++b}, s_{+-b}, s_{--b}, s_{-+b}, s_{++b}, s_{+-b}, s_{--b}, s_{-+b})\}$

## D.4. Computational Cost and Complexity

As mentioned in Section 7, training the composition map is the most time-consuming process, as it requires a preprocessing step of fitting every admissible composition to each sample (this can be parallelized). The actual time to fit the decision tree is negligible in our experiments. This preprocessing step has time complexity $O(DM|\mathcal{C}'|)$ where $D$ is the number of samples, $M$ is the dimensionality of the trajectory, and $|\mathcal{C}'|$ is the number of admissible compositions. Crucially, this time does not depend on $K$ (the dimensionality of static features $\mathcal{V}$).

Table 8 shows the time needed to train the composition map for each dataset. We currently use a computationally intensive procedure of minimizing a Dynamic Time Warping (DTW) distance and then the standard MSE error. There is a trade-off between the accuracy of fit and the computation time. Practitioners may opt for more approximate fits (e.g., by not using DTW).

*Table 8.* Time needed to train the composition map for each of the datasets. All experiments were performed on 18-core Intel Core i9-10980XE with 60GB of RAM.

| Dataset | Noise | Time (min) | # compositions | # samples | Time per sample (s) | Time per composition (s) | Time per single fit (s) |
|---|---|---|---|---|---|---|---|
| SIR | 0.01 | 233.67 | 26 | 500 | 28.04 | 539.24 | 0.36 |
| SIR | 0.1 | 178.66 | 26 | 500 | 21.44 | 412.28 | 0.27 |
| PK | 0.01 | 68.73 | 26 | 500 | 8.25 | 158.60 | 0.32 |
| PK | 0.1 | 66.65 | 26 | 500 | 8.00 | 153.80 | 0.31 |
| Tumor | 0.01 | 77.33 | 26 | 500 | 9.28 | 178.45 | 0.36 |
| Tumor | 0.1 | 66.37 | 26 | 500 | 7.96 | 153.16 | 0.31 |
| Tacrolimus | real | 8.65 | 26 | 90 | 5.77 | 19.96 | 0.22 |
| Bike sharing | real | 462.27 | 34 | 655 | 42.35 | 815.77 | 1.25 |
| HIV | 0.01 | 259.33 | 26 | 500 | 31.12 | 598.44 | 0.40 |
| HIV | 0.1 | 194.33 | 26 | 500 | 23.32 | 448.45 | 0.30 |

The advantage of training the composition map separately from the property map is that the errors table $e^{(d)}[c]$ (Equation (1)) can be computed once, saved, and then reused for different training settings of the composition decision tree and the property maps. This makes introducing changes to the model much quicker.

Table 9 shows the overall computational cost for all methods on all datasets (including five runs and hyperparameter tuning). Note that we calculate the errors table $e^{(d)}[c]$ (Equation (1)) once for all five seeds and then subsample the results based on the split.

*Table 9.* Computation time (in minutes) for all experiments (five runs with hyperparameter tuning). All experiments were performed on 18-core Intel Core i9-10980XE with 60GB of RAM.

| | SIR | | PK | | Tumor | | Tacrolimus | Bike sharing | HIV | |
| --- | --- | --- | --- | --- | --- | --- | --- | --- | --- | --- |
| Noise | $\sigma = 0.01$ | $\sigma = 0.1$ | $\sigma = 0.01$ | $\sigma = 0.1$ | $\sigma = 0.01$ | $\sigma = 0.1$ | real | real | $\sigma = 0.01$ | $\sigma = 0.1$ |
| NeuralODE | 86.07 | 215.03 | 94.09 | 521.42 | 49.90 | 56.70 | 9.57 | 146.17 | 265.32 | 152.75 |
| ANODE | 94.79 | 185.77 | 94.36 | 291.19 | 56.09 | 62.32 | 10.95 | 187.07 | 139.76 | 140.12 |
| LatentODE | 67.63 | 149.08 | 56.64 | 64.77 | 46.66 | 51.05 | 6.92 | 94.78 | 79.56 | 57.19 |
| SINDy-5 | 14.87 | 26.11 | 40.91 | 44.43 | 8.96 | 23.39 | 14.03 | 25.22 | 81.38 | 91.08 |
| SINDy-20 | 15.99 | 23.21 | 57.63 | 82.76 | 8.19 | 19.51 | 32.68 | 60.86 | 105.65 | 93.98 |
| WSINDy-5 | 12.74 | 13.58 | 26.41 | 13.00 | 10.83 | 11.16 | 32.06 | 15.62 | 42.56 | 29.51 |
| WSINDy-20 | 11.60 | 13.36 | 44.76 | 115.38 | 10.05 | 10.39 | 38.83 | 39.06 | 42.04 | 27.96 |
| EPISODE (property maps) | 100.20 | 75.64 | 118.07 | 44.96 | 31.62 | 55.84 | 35.12 | 104.84 | 166.23 | 169.67 |
| EPISODE (composition maps) | 233.67 | 178.66 | 68.73 | 66.65 | 77.33 | 66.37 | 8.65 | 462.27 | 259.33 | 194.33 |
| EPISODE (whole) | 333.87 | 254.30 | 186.80 | 111.61 | 108.95 | 122.21 | 43.77 | 567.11 | 425.55 | 364.00 |
| EPISODE* (property maps) | 98.04 | 82.87 | 55.15 | 53.35 | 23.02 | 24.42 | 34.34 | 85.57 | 152.04 | 156.83 |
| EPISODE* (composition maps) | 8.99 | 6.87 | 2.64 | 2.56 | 11.90 | 10.21 | 0.33 | 27.19 | 26.60 | 19.93 |
| EPISODE* (whole) | 107.03 | 89.74 | 57.79 | 55.91 | 34.92 | 34.64 | 34.67 | 112.76 | 178.63 | 176.76 |

# E. Extended Related Works

**Symbolic Regression and Discovery of Differential Equations** The identification of differential equations is often viewed as part of the larger field known as symbolic regression. This area of machine learning aims to express data through a closed-form mathematical expression. Traditionally, genetic programming has been employed for symbolic regression (Stephens, 2022; Cranmer, 2020), but recent advancements have seen the integration of neural networks. This involves methods such as directly representing equations within neural networks by modifying activation functions (Martius & Lampert, 2017; Sahoo et al., 2018), leveraging neural networks to narrow down the search space (Udrescu & Tegmark, 2020; Udrescu et al., 2021), exploring equations through reinforcement learning (Petersen et al., 2021), and employing large pre-trained transformers (Biggio et al., 2021; D'Ascoli et al., 2022; Holt et al., 2023). Standard symbolic regression can be adapted for ordinary differential equation (ODE) discovery by estimating derivatives from data to treat them as targets (Quade et al., 2016). Despite this, several specialized ODE discovery techniques have emerged. Among these, the Sparse Identification of Nonlinear Dynamical Systems (SINDy) (Brunton et al., 2016b) stands out, describing derivatives as linear combinations of functions from a predetermined library. This has led to various extensions, including those for implicit equations (Kaheman et al., 2020), controlled equations (Brunton et al., 2016b), and treatment effect estimation (Kacprzyk et al., 2024a). Alternative approaches utilize weak formulations of ODEs, circumventing the need for derivative estimation (Messenger & Bortz, 2021a; Qian et al., 2022). Additionally, methods specifically designed for discovering partial differential equations (PDEs) have been developed, as standard ODE discovery methods cannot be directly applied to PDEs (Rudy et al., 2017; Raissi & Karniadakis, 2018; Messenger & Bortz, 2021b; Kacprzyk et al., 2023). The challenge of finding comprehensible and well-fitting closed-form expressions inspired Shape Arithmetic Expressions (Kacprzyk & van der Schaar, 2024) that extend the prespecified set of well-known functions (e.g., trigonometric or exponential functions) in symbolic regression by flexible and learnable univariate functions. Those functions do not necessarily have a compact symbolic representation but are supposed to be understood by looking at their line plot.

**Black Box Approaches for Modeling Dynamical Systems** Black-box approaches to modeling dynamical systems have gained significant attention due to their ability to learn complex dynamics without requiring explicit governing equations. Neural Ordinary Differential Equations (Neural ODEs) (Chen et al., 2018) introduced a model that parameterizes differential equations using neural networks, enabling flexible trajectory modeling. Building upon this, Augmented Neural ODEs (Dupont et al., 2019) addresses the limitations of Neural ODEs in handling complex dynamics by expanding the state space, allowing for more expressive representations. Latent ODEs (Rubanova et al., 2019) further extend this framework by inferring latent representations of time-series data. Neural Laplace (Holt et al., 2022) generalizes Neural ODEs by learning solutions in the Laplace domain, offering improved performance for systems governed by diverse types of differential equations.

# F. Additional Discussion

### F.1. Sensitivity to the Choice of $\mathcal{C}'$

The results in Table 2 (EPISODE) are for a generous set of compositions. We consider all compositions up to length 4 (or length 8 for the Bike sharing dataset) except for compositions where two consecutive transition points are inflection points. This results in 26 and 34 compositions, respectively. Thus, EPISODE achieves competitive results for a relatively large choice of compositions (and minimum prior knowledge). EPISODE* uses prior knowledge about the shape of the compositions, constraining the set to between 1 and 4 compositions depending on the dataset (details in Appendix D.3). We can see that it significantly improves the performance in some settings (Tumor dataset), but often has no significant impact (Bike sharing, Tacrolimus, HIV). In these settings, EPISODE is able to autonomously determine which compositions should be used in the composition map.

### F.2. Extending to Periodic or Oscillating Trajectories

EPISODE could, in the future, be extended to periodic or oscillating trajectories by extending the definition of semantic representation to accommodate periodic compositions. Let us assume that we know that the trajectories are described by an infinite composition consisting of repeating $(s_{+-c}, s_{--c}, s_{-+c}, s_{++c})$. Then we can describe this composition segment using properties like "amplitude" or "frequency". For each trajectory in our training dataset, we can extract additional trajectories describing how these properties change over time. We can now model these auxiliary trajectories using a framework similar to EPISODE. However, we leave a full exposition of this idea for a future paper.

