# OpenReview forum: "Skip the Equations: Learning Behavior of Personalized Dynamical Systems Directly From Data"
_ICML.cc/2025/Conference — ICML 2025 poster_

### Official Review · Reviewer_htLB · 2025-03-11

**Overall Recommendation:** 3

**Summary:**

The paper tackles the modeling of personalized dynamical systems — that is, dynamical systems whose trajectory evolves conditioned on a set of static parameters, such as the initial condition in ordinary differential equation (ODE) systems, together with other "personal" covariates, such as one patient's weight or age, which affect the dynamics of e.g. drug metabolism in the patient. To do this, the paper extends the very recent Semantic ODE framework of Kacprzyk and van der Schaar (2025), which break down the shape of ODE trajectories into a *composition* (a sequence of *motifs*  such as "increasing and convex" or "decreasing and concave") and their *properties* (the specifics of these motifs, such as the locations and heights of local maxima).

Semantic ODE is trained to predict the easily interpretable semantic composition and its properties that best describe one-dimensional noisy trajectories, given the initial condition of the system. The authors extend Semantic ODEs in two directions:

1. First, they allow for the modeling of systems of dimension larger than one, by making use of a  "channel independent strategy" — that is, by treating each component of the target multidimensional system as independent.
2. Second, they allow the composition map and properties sub-maps to process multidimensional covariates.

The authors test their methodology in different synthetic and real world datasets that correspond to very different, personalized dynamical systems, and compare their performance against that of symbolic and neural ODE models.

*References*:
- Kacprzyk and van der Schaar (2025): No Equations Needed: Learning System Dynamics Without Relying on Closed-Form ODEs, ICLR

## Update After Review

The authors' replies to my questions and concerns were satisfactory. Hence, I increased my score.

**Claims And Evidence:**

The authors claim that their methodology enables practitioners to:

1. readily integrate prior knowledge, which is done via the pre-selection of the composition library (as in Section 5.2);
2. easily understand the main aspect of the dynamics, through the interpretability of their motif (as in Section 5.3);
3. ensure desired behavior and revise the model when necessary (as in Section 5.4 and 5.5).

The authors also claim that their model is flexible enough to be competitive in forecasting tasks.

Overall, their results in Sections 5 and 6 align well with these claims. However, **most of these claims also apply to the original Semantic ODE framework on which this work is based**.

The proposal extends Semantic ODE to multidimensional systems with multidimensional covariates. **Although their target datasets do feature different covariates, they mostly consists of one-dimensional processes**. Furthermore, the authors never display their target datasets nor their predictions, so one can only rely on the scores in Tables 1 and 2, **which are also not described**.

**Essential References Not Discussed:**

**I do not know about any fundamental paper missed by the authors**. On the contrary, the authors  compared their methodology against different ODE-based models, which are widely used by the dynamical system and machine learning community. What is more, since the proposal's most crucial aspects are its transparency, verifiability, and editability (as written by the authors), I do not think it is necessary to compare their method with the most recent, transformer-based forecasting models that do not rely on continuous-time representations.

**Experimental Designs Or Analyses:**

The organization of Section 5 nicely illustrates how the proposal addresses all the claims made in the introduction. Section 6 compares the proposal against well-established ODE-based methods, and Appendix D contains the details about the target datasets and the training details of all baselines.

Unfortunately, and as I wrote above, there is little to no information regarding the metrics reported in Table 1 and 2.

**Methods And Evaluation Criteria:**

The methodology proposed by the authors is interesting. Their results demonstrate that it represents an alternative to ODE modeling which, as the author point out, does not necessarily lead to easily interpretable representations of the dynamics.

*Regarding evaluation*, I do not manage to find what metric is reported in Tables 1 and 2.  From the introduction one gathers that it must correspond to some divergence between target and predictions up to some horizon time (i.e. a forecasting task). **None of these details are provided** (or at least, I could not find them).

**Other Comments Or Suggestions:**

On the second column, first paragraph of page 4 the authors use both $v \in \mathcal{V}$ and $\boldsymbol{v} \in \mathcal{V}$ to refer to the covariates. Unless I am misunderstanding something, they always correspond to vectors, right?

**Other Strengths And Weaknesses:**

I think the main weakness I find is that the paper could be seen as incremental, for it relies too much on the Semantic ODE framework of Kacprzyk and van der Schaar (2025).

Let me summarize the main weaknesses I highlight above:
1. The authors only study one target dataset of dimension larger than one. This is a weakness because one of the main claims of the authors is that they extend Semantic ODE to multidimensional systems.
2. The specifics of the trajectory predictor $F_{traj}$ are missing. The paper shoul be self-contained.
3. The introduction borrows many lines from Kacprzyk and van der Schaar (2025).
4. The authors do not display the target trajectories nor their predictions. The time series forecasting community often includes such plots, because they help readers better understand the performance of the models under study. It would also illustrate the connection between the motifs predicted by the model and the actual data.
5. There is no description of the metrics used to compute the scores in Tables 1 and 2 (or at least, I did not find them).
6. The authors give no details (or I did not find them) regarding the computational cost of their proposal. For example, how long does it take to optimize their models on the different datasets as compared to the baselines?

**Questions For Authors:**

1. How far into the future you predict the dynamics of the systems you study?
2. The strength of the noise corruption is set to 0.01. How does the model respond to different noise corruptions? Is it as robust as Semantic ODE?

**Relation To Broader Scientific Literature:**

The authors heavily rely on the work by Kacprzyk and van der Schaar (2025), which introduced Semantic ODE. Indeed, the proposal extends Semantic ODE to multidimensional systems with several static covariates, *which required novel implementation of the composition map and properties sub-maps*. These extensions are important and have great applicability.

However, only one target dataset has dimension larger than one. The paper would benefit from experiments in other higher dimensional systems — even if such system will often display periodic or periodic-like behavior, which is hard to model with the proposed methodology, as acknowledged by the authors.

Furthermore, there are some details, like the specifics of the trajectory predictor $F_{traj}$, which the authors could include into the Appendix. Otherwise the reader is forced to read Kacprzyk and van der Schaar (2025) to understand the proposal. In my opinion, an ICML paper should be self-contained. Specially given that the work by Kacprzyk and van der Schaar (2025) just came out.

Finally, the introduction section borrows many lines from Kacprzyk and van der Schaar (2025). This is to be avoided.

**Theoretical Claims:**

There are no major theoretical claims in this work.

---

> ### Author Rebuttal · Authors · 2025-04-01
>
> Dear Reviewer htLB,
>
> Thank you so much for such a comprehensive review. We deeply appreciate your time and attention spent on our paper. We are glad that you found our methodology interesting, comparisons comprehensive, and claims justified. We answer the six weaknesses in your summary, then the two questions. Finally, we address your comment about the incremental nature of our work.
>
> ### W1. Datasets with more than one dimension
> We have now added experiments on the viral dynamics model of HIV [1] ($M=3$) in two noise settings, where the features include not only the initial conditions but also four additional parameters. We have also tested our method on the SIR model with more noise. Overall, we have added three additional multidimensional datasets (all results can be seen [here](https://imgur.com/a/5nnGzqi).
>
> The main challenge of extending Semantic ODE to multidimensional systems lies in the fact that these trajectories no longer can depend on a single initial condition $x_0\in\mathbb{R}$ but on a multidimensional $\mathbf{x}_0 \in \mathbb{R}^M$ (where $M>1$). Thus, the main contribution is the extension to multidimensional inputs, which we demonstrate in all our experiments.
>
> We hope that this explanation and the additional experiments provide sufficient evidence for our claim.
>
> **Actions taken**: Added additional experiments on multidimensional datasets in Appendix A.
>
> ### W2. Details of $F_{\text{traj}}$
> **Actions taken**: Added description of the trajectory predictor in Appendix E as well as details on compositions, motifs, and properties to make the paper self-contained.
>
> ### W3. Similar examples in the introduction
> **Actions taken**: Replaced some of the examples and references in the introduction to limit overlap with the paper on Semantic ODEs.
>
> ### W4. Predicted trajectories
> **Actions taken**: Included plots of sample trajectories predicted by EPISODE in Appendix A. These can be seen [here](https://imgur.com/a/IZHZ6l2).
>
> ### W5. Metrics used
> Please see our response to Reviewer g6tH (Evaluation metric).
>
> **Actions taken**: Described the used metric in Section 5.
>
> ### W6. Computational cost
>
> We have now added a [table](https://imgur.com/a/4jhWn1H) to Appendix D showing the overall computational cost of all approaches (including five runs and hyperparameter tuning). We also show the individual times to train the composition map and the property maps. All experiments were performed on an 18-core Intel Core i9-10980XE with 60GB of RAM.
>
> As mentioned in the limitations, training of the composition map is the most time-consuming process as it requires fitting every admissible composition to each sample (this can be parallelized). Note that we perform this preprocessing *once* for all five seeds and then subsample the results based on the split. The actual time to fit the decision tree is negligible. A more detailed breakdown of the time needed to train the composition map is provided in the [following table](https://imgur.com/a/90rECtR). We discuss it further in our response to Reviewer evCm (Q1).
>
> **Actions taken:** Added computation times in Appendix D.
>
> ### Q1 Time horizon
> Each dataset has a specific time horizon that is subsequently scaled to $(0,1)$. The details are in Appendix D.1. However, for unbounded motifs, our model can predict for any time $t\geq0$. We evaluate each model on a on a held-out dataset of samples, whose trajectories are observed at the same time points as the ones in the training dataset.
>
> ### Q2 Robustness
> **Actions taken**: Added additional five experiments on datasets with higher noise ($\sigma=0.1$) in Appendix A. EPISODE shows robustness to noise in these settings. The results can be seen [here](https://imgur.com/a/3pjzwQS)
>
> ### Minor
> **Actions taken**: Fixed the typo on page 4 (it should have been $\mathbf{v} \in \mathcal{V}$).
>
> ### Incremental contributions
>
> Semantic ODE only works for one-dimensional trajectories predicted from one-dimensional inputs. We have extended it to more realistic settings. In particular, we
> 1. Extended the composition map from interval partition to a decision tree and designed a novel optimization algorithm for fitting it.
> 2. Extended the property functions from univariate functions to GAMs and designed a novel optimization algorithm for fitting GAMs such that the predicted properties are consistent with the composition.
> 3. Improved composition map training by using Dynamic Time Warping distance.
> 4. Accommodated bounded compositions ($t_{\text{end}}\in\mathbb{R}$).
> 5. Accommodated categorical variables as features.
> 6. Implemented a visualization tool to visualize the fitted GAMs.
>
> **References**
>
> 1. Hill, A. L., Rosenbloom, D. I., Nowak, M. A., & Siliciano, R. F. (2018). Insight into treatment of HIV infection from viral dynamics models. Immunological reviews, 285(1), 9-25.
>
> ---
>
> We hope we have addressed all your concerns. Please let us know if you have any additional questions, we are eager to address them.
>
> Kind regards,
>
> Authors

---

> > ### Comment · Reviewer_htLB · 2025-04-04
> >
> > Dear authors,
> >
> > thanks very much for your replies. I have increased my score.

---

> > > ### Author Response · Authors · 2025-04-09
> > >
> > > Dear Reviewer htLB,
> > >
> > > Thank you very much for your reply and for increasing your score. Your comments helped us improve our paper by adding clarifications, experimental details, additional datasets, and noise settings. We hope to have the opportunity to present our work at ICML, where we believe it can advance data-driven modeling of dynamical systems beyond black-box methods and closed-form expressions.
> > >
> > > Kind regards,
> > >
> > > Authors

---

### Official Review · Reviewer_g6tH · 2025-03-11

**Overall Recommendation:** 2

**Summary:**

The paper proposes EPISODE, a framework for learning the behavior of personalized dynamical systems without requiring explicit equation discovery. Instead of the traditional two-step approach of identifying ODEs and then analyzing them, EPISODE directly predicts the semantic representation from data. The paper extends prior direct semantic modeling approaches to accommodate multi-dimensional trajectories and integrate personalization via auxiliary static features.

**Claims And Evidence:**

The paper claims to provide an alternative to equation-based modeling by learning behavior directly from data, which is supported by a detailed introduction to the algorithm and experimental results on both synthetic and real-world datasets.

**Essential References Not Discussed:**

NAN

**Experimental Designs Or Analyses:**

The case study on Tacrolimus is well-motivated. The comparison with different model classes is comprehensive. However, the computation efficiency are not clearly discussed.

**Methods And Evaluation Criteria:**

The method seems reasonable but highly relies on the incorporation of semantic inductive biases. Such incorporation benefits the learning process but, at the same time, depends on prior knowledge about the behavior of the dynamical system, raising questions about its scalability to complex systems.

Additionally, the realization is not end-to-end, requiring manual verification and editing, which raises concerns about efficiency.

The performance comparison is first mentioned in line 420, but I did not find an introduction to the evaluation metric. Moreover, based on the results shown in Tables 1 and 2, the method does not significantly outperform other baselines.

**Other Comments Or Suggestions:**

See above.

**Other Strengths And Weaknesses:**

See above.

**Questions For Authors:**

See above.

**Relation To Broader Scientific Literature:**

The work is closely related to recent efforts in physics-informed machine learning and symbolic regression.

**Theoretical Claims:**

The paper provides a solid conceptual framework, and no obvious issues are found with the theorem.

---

> ### Author Rebuttal · Authors · 2025-04-01
>
> Dear Reviewer g6tH,
>
> Thank you very much for your review. We appreciate your time and effort spent reviewing our paper. We are glad you found our conceptual framework solid, the case study well-motivated, and the comparisons comprehensive. We address your comments below.
>
> ### Reliance on inductive biases
>
> > The method seems reasonable but highly relies on the incorporation of semantic inductive biases.
>
> Thank you for this comment, as it allows us to clarify the experimental setup in our paper. Although our method *can* incorporate semantic inductive biases, it *does not require* this information. The results in Table 2 (EPISODE) are for a generous set of compositions without much prior knowledge. We consider all compositions up to length 4 (or length 8 for the Bike sharing dataset) except for compositions where two consecutive transition points are inflection points. This ends up being, respectively, 26 and 34 compositions. Thus, EPISODE achieves competitive results for a relatively large choice of compositions and minimum prior knowledge. Often, the results are not significantly different for EPISODE*, where we incorporate such knowledge (see Tacrolimus or Bike sharing dataset). Prior knowledge is, however, useful in reducing the training time, satisfying requirements, and improving extrapolation.
>
> **Actions taken**: We have now clarified the amount of prior knowledge used in experiments in Section 6.
>
> ### End-to-end
>
> > Additionally, the realization is not end-to-end, requiring manual verification and editing ...
>
> We would like to clarify that, although manual verification and editing are big advantages of our approach, they are *not necessary*. Although the training process consists of two steps (training the composition map and then the property maps), the whole pipeline *is* end-to-end. There is *no manual intervention* in experiments performed in Section 6.
>
> **Actions taken**: We have now clarified the absence of manual intervention in experiments in Section 6.
>
> ### Evaluation metric
>
> Thank you for pointing this out! We have now described the used metric in Sections 5 and 6. The error is given by
>
> $$ \mathcal{L} = \frac{1}{D} \sum_{d=1}^D \sqrt{\frac{1}{N_d} \sum_{n=1}^{N_d} \left(\frac{1}{M} ||\mathbf{F}(\mathbf{v}^{(d)})(t_n^{(d)})-\mathbf{y}_n^{(d)}||_2^2\right)} $$
>
> where $\mathbf{F}$ is the predictive model (EPISODE), $D$ is the number of samples, $M$ is the dimensionality of the system, $N_d$ is the number of measurements of the $d^{\text{th}}$ sample $(\mathbf{v}^{(d)}, (t\_n^{(d)}, \mathbf{y}\_n^{(d)})\_{n=1}^{N\_d})$. We choose this metric because for $M=1$ it reduces to the standard mean RMSE, i.e., $\frac{1}{D}\sum_{d=1}^D \sqrt{\frac{1}{N_d} \sum_{n=1}^{N_d} (F(\mathbf{v}^{(d)})(t_n^{(d)})-y_n^{(d)})^2}$  (used in Semantic ODE paper) and we normalize by $M$ so that it is easier to compare results between systems of different dimensionality.
>
> The error is calculated on a held-out dataset of samples, whose trajectories are observed at the same time points as the ones in the training dataset.
>
> **Actions taken**: Described the used metric in Sections 5 and 6.
>
> ### Computational cost
>
> We have now added a [table](https://imgur.com/a/4jhWn1H) to Appendix D showing the overall computational cost of all approaches (including five runs and hyperparameter tuning). We also show the individual times to train the composition map and the property maps. All experiments were performed on an 18-core Intel Core i9-10980XE with 60GB of RAM.
>
> As mentioned in the limitations, training of the composition map is the most time-consuming process as it requires fitting every admissible composition to each sample (this can be parallelized). Note that we perform this preprocessing *once* for all five seeds and then subsample the results based on the split. A more detailed breakdown of the time needed to train the composition map is provided in the [following table](https://imgur.com/a/90rECtR). We discuss it further in our response to Reviewer evCm (Q1).
>
> **Actions taken:** Added computation times in Appendix D.
>
> ### Performance
>
> We have run additional experiments, including a new dataset with the viral dynamics model of HIV and settings with higher noise. All of the results can be seen [here](https://imgur.com/a/5nnGzqi).
>
> EPISODE* achieves nearly perfect performance on the SIR dataset, demonstrating superior noise robustness to SINDy whose constrained search space should have given it an advantage on this simple problem. Both variants of EPISODE show superior performance on the PK and HIV datasets (in both low and high noise settings) compared to both ODE discovery and black-box models. Both variants also achieve superior performance to ODE discovery methods on the two real datasets (Tacrolimus and Bike sharing).
>
> ---
>
> We hope we have addressed all your concerns. Please let us know if you have any additional questions, we are more than happy to address them.
>
> Kind regards,
>
> Authors

---

### Official Review · Reviewer_evCm · 2025-03-12

**Overall Recommendation:** 4

**Summary:**

The paper proposes a method called EPISODE for learning personalized dynamical systems (PDS) without explicitly discovering ordinary differential equations (ODEs). As mentioned in this paper, traditional approaches to modeling dynamical systems involve first identifying closed-form equations and then analyzing their properties. In contrast, EPISODE bypasses this two-step process by directly predicting a semantic representation of the system’s behavior (including the trajectory’s shape and key quantitative properties) from static inputs or covariates. The method utilizes decision trees to predict the shape of trajectories and generalized additive models (GAMs) to model their quantitative properties.

## update after rebuttal
I am quite satisfied with the rebuttal.

**Claims And Evidence:**

The paper generally supports its claims with clear evidence. The key claim is that EPISODE offers a transparent, editable, and interpretable alternative to traditional equation discovery or black-box methods. This claim is supported by three aspects:
- Demonstrations on synthetic and real datasets (e.g., SIR model, Tacrolimus).
- Comparisons against established baselines.
- The detailed case study shows the practical benefit of transparency, interpretability, and editability.
However, some claims regarding scalability to more complex dynamical systems or handling oscillatory behaviors are acknowledged as limitations.

**Essential References Not Discussed:**

The paper adequately references and discusses existing works central to understanding its contribution. No immediate essential references seem missing.

**Ethical Review Concerns:**

N.A.

**Experimental Designs Or Analyses:**

Yes. The experimental designs and analyses are sound and clearly structured. Experiments effectively illustrate the method’s strengths in interpretability and performance. The authors also conduct clear ablation studies and robustness checks.

**Methods And Evaluation Criteria:**

In my opinion, the methods and evaluation criteria are very reasonable and relevant. EPISODE is rigorously compared with both closed-form ODE discovery methods (SINDy variants) and black-box approaches (Neural ODEs variants), using metrics such as mean squared error (MSE). The authors also carefully incorporate practical interpretability considerations into their evaluation, demonstrating the method’s practical benefits through real-world medical datasets.

**Other Comments Or Suggestions:**

The writing and presentation are generally clear. Minor typos (e.g., spacing, punctuation) could be corrected upon careful proofreading. More explicit guidelines on interpreting GAM plots might further benefit readers unfamiliar with GAMs.

**Other Strengths And Weaknesses:**

Strengths:
- Novel combination of direct semantic modeling and GAMs, resulting in transparent and interpretable modeling. Simple but effective.
- Practical relevance demonstrated through pharmacokinetic applications.


Weaknesses:
- Method restricted to finite compositions, limiting application to oscillatory or periodic systems.
- The complexity and runtime involved in the composition map training could limit scalability.

**Questions For Authors:**

1. What is the runtime complexity of the composition map training as the dataset size and dimensionality grow?
2. How sensitive is the method to the initial choice of compositions?
3. Could an automatic or semi-automatic strategy for composition selection apply?
4. How to possibly adapt or extend EPISODE to systems with periodic or oscillatory behavior?

**Relation To Broader Scientific Literature:**

The paper positions itself clearly within the broader literature on dynamical system modeling and equation discovery (e.g., references to SINDy, Neural ODEs, ANODE, Semantic ODEs).

**Theoretical Claims:**

Yes. But the paper does not primarily rely on complex theoretical claims.

---

> ### Author Rebuttal · Authors · 2025-04-01
>
> Dear Reviewer evCm,
>
> Thank you very much for your positive review! We are glad you found our method novel and our claims well-evidenced. We reply to your questions first, and then we discuss other weaknesses and suggestions.
>
> ### Q1. Runtime complexity of the composition map training
>
> As mentioned in the limitations, training the composition map is the most time-consuming process as it requires a preprocessing step of fitting every admissible composition to each sample (this can be parallelized). The actual time to fit the decision tree is negligible in our experiments. This preprocessing step has time complexity $O(D M |\mathcal{C}'|)$ where $D$ is the number of samples, $M$ is the dimensionality of the trajectory, and $|\mathcal{C}'|$ is the number of admissible compositions. Crucially, this time does not depend on $K$ (the dimensionality of static features $\mathcal{V}$).
>
> We have now included a table showing the time needed to train the composition map for each dataset (see [here](https://imgur.com/a/90rECtR)). We currently use a computationally intensive procedure of minimizing a Dynamic Time Warping (DTW) distance and then the standard MSE error. There is a trade-off between the accuracy of fit and the computation time. Practitioners may opt for more approximate fits (e.g., by not using DTW).
>
> The advantage of training the composition map separately from the property map is that the errors table $e^{(d)}[c]$ (Eq. 1) can be computed once, saved, and then reused for different training settings of the composition decision tree and the property maps. This makes introducing changes to the model much quicker. The training time of the property maps (for five seeds and with tuning) can be found [here](https://imgur.com/a/4jhWn1H).
>
> **Actions taken**: Added a discussion on the computational complexity in a newly created Appendix F (Additional Discussion) and tables with computation times in Appendix D.
>
>
> ### Q2. Sensitivity to the initial choice of compositions
>
> The results in Table 2 (EPISODE) are for a generous set of compositions. We consider all compositions up to length 4 (or length 8 for the Bike sharing dataset) except for compositions where two consecutive transition points are inflection points. This ends up being, respectively, 26 and 34 compositions. Thus, EPISODE achieves competitive results for a relatively large choice of compositions (and minimum prior knowledge). EPISODE* uses prior knowledge about the shape of the compositions, constraining the set to between 1 and 4 compositions depending on the dataset (details in Appendix D.3). We can see that it significantly improves the performance in some settings (Tumor dataset) but often has no significant impact (Bike sharing dataset or Tacrolimus). In these settings, EPISODE is able to autonomously determine which compositions should be used in the composition map. This finding generalizes to the additional experiments that we have run (see [here](https://imgur.com/a/5nnGzqi)).
>
> **Actions taken**: Added a discussion on the sensitivity of the method to the choice of the set of admissible compositions in a newly created Appendix F (Additional Discussion).
>
> ### Q3. Automatic strategy for composition selection
>
> As described above, the algorithm can usually autonomously determine which compositions should be used in the composition map. Thus, a viable approach is to start with a large number of compositions and let the algorithm narrow it down. In the future, we would like to design a fast algorithm that provides an approximate fit between a composition and a trajectory that narrows down the search space before the actual fitting.
>
> ### Q4. Periodic or oscillatory behavior
>
> EPISODE could, in the future, be extended to periodic or oscillating trajectories by extending the definition of semantic representation to accommodate periodic compositions. Let us assume that we know that the trajectories are described by an infinite composition consisting of repeating $(s_{+-c},s_{--c},s_{-+c},s_{++c})$. Then we can describe this composition segment using properties like "amplitude" or "frequency". For each trajectory in our training dataset we can extract additional trajectories describing how these properties change over time. We can now model these auxiliary trajectories using a framework similar to EPISODE. However, we leave a full exposition of this idea for a future paper.
>
> **Actions taken**: Added a discussion on extending EPISODE to infinite compositions in a newly created Appendix F (Additional Discussion).
>
> ### W1. Finite compositions
>
> Please see our response to Q4.
>
> ### W2. Complexity of composition map training
>
> Please see our response to Q1.
>
> ### Minor
> **Actions taken**: We have now added explicit guidelines on interpreting GAM plots in Appendix E.
>
>
> ---
>
> We hope we have addressed all your concerns. Please let us know if you have any additional questions, we are more than happy to address them.
>
> Kind regards,
>
> Authors

---

### Decision · Program_Chairs · 2025-05-01

**Decision:**

Accept (poster)

**Comment:**

This manuscript presents a method to identify personalized dynamics through a compositional set of motifs that remove the need for formulating parametrized closed-form equations that are then adapted to each individual. The work is based on decision trees for the compositional maps and generalized additive models. Key to the motivation is the ability to work on multi-dimensional systems (as opposed to prior work in this area that has focused on 1D time series). This is accomplished through separating and treating each dimension separately.

While the reviewers made note of some weaknesses, such as the potential to scale and limitations in some cases, e.g., periodic systems, these were viewed as minor weaknesses and the authors were able to respond to most of them. I therefore believe that this work would be a good addition to the ICML conference.